# HIERARCHICAL EQUIVARIANT GRAPH GENERATION

## ABSTRACT

Deep learning, and more specifically denoising models, have significantly improved graph generative modeling. However, challenges remain in capturing global graph properties from local interactions, ensuring scalability, and maintaining node permutation equivariance. While existing equivariant models address node permutation issues, they struggle with scalability, often requiring dense graph representations that scale with $\mathcal{O}(n^2)$.

To overcome these challenges, we introduce a novel coarsening-lifting method that generates sparse spanning supergraphs, preserving global graph properties. These supergraphs serve as both conditioning structures and sparse message-passing layouts for generative models. Leveraging this method with discrete diffusion, we model graphs hierarchically, enabling efficient generation of large graphs.

Our approach, to the best of our knowledge, is the first hierarchical equivariant generative model for graphs. We demonstrate its performance introducing new evaluation datasets with larger graphs and more instances than traditional benchmarks.

## 1 INTRODUCTION

Graph generation is a research area with important applications, including drug discovery, material design (Lu et al., 2020), protein design (Ingraham et al., 2019), programming code modeling (Brockschmidt et al., 2019), natural language processing (Chen et al., 2018; Klawonn & Heim, 2018), and robotics (Li et al., 2017). Advancements in deep generative models, such as deep autoregressive models, generative adversarial networks, variational auto-encoders, and vector-quantized auto-encoders, have significantly benefited graph modeling. Recently, the adoption of denoising models (Jo et al., 2022; Yang et al., 2019; Haefeli et al., 2022; Jo et al., 2024; Qin et al., 2024; Eijkelboom et al., 2024) has further improved graph distribution modeling.

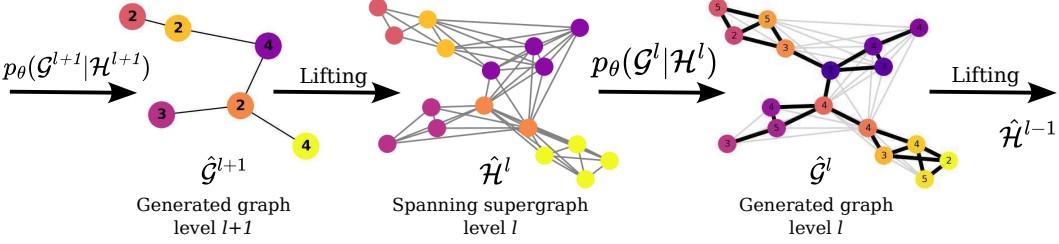

Figure 1: One step of generative process: At level $l$, we receive the generated coarse graph $\hat{\mathcal{G}}^{l+1}$ from the coarseness level $l + 1$, which we lift to obtain the conditioning graph $\mathcal{H}^l$ (colors and annotation in $\hat{\mathcal{G}}^{l+1}$ correspond to colors and number of node in $\mathcal{H}^l$). Using the learned conditional distribution $p_\theta(\mathcal{G}^l|\mathcal{H}^l)$, we sample $\hat{\mathcal{G}}^l$ (represented in bold black edges) in the graph space defined by $\mathcal{H}^l$ (light gray edges, which implies that $\mathcal{H}^l$ is a spanning supergraph of $\mathcal{G}^l$). In its turn, $\hat{\mathcal{G}}^l$ is lifted and passed to the finer level $l - 1$. We iterate this process until we reach the original data level, i.e. level 0.

Despite these advancements, generative graph modeling still faces several specific and challenging issues such as the multiple possible representations due to node permutations, the difficulty in capturing global graph properties from local node interactions, and scalability concerns.

Models equivariant to node permutations (referred to concisely as *equivariant models*) offer an elegant solution to the issue of multiple representations by ensuring a unique computational graph for all possible instantiations of the same graph. Empirical evidence suggests that equivariant models better capture the distribution of small graphs compared to their non-equivariant counterparts (Jo et al., 2022). However, in the absence of prior information and generation order, equivariant models need to consider all node pairs in parallel, which exacerbates issues related to capturing global graph properties and scalability. For that reason, most equivariant models use a dense graph representation (Haefeli et al., 2022; Vignac et al., 2023; Jo et al., 2022; Boget et al., 2024). A few equivariant models (Qin et al., 2024; Chen et al., 2023) adopt a sparse representation by focusing on a subset of possible node pairs at any given time; however, they still ultimately need to account for all node pairs. Unlike some models for molecule generation that construct molecular graphs based on 3D atomic coordinates (Hoogeboom et al., 2022; Xu et al., 2023), our work focuses on the task of general graph generation.

We propose a method to address these two issues while preserving equivariance. Our approach generates coarse graphs, which are expanded into spanning supergraphs that capture spectral properties of the graph and provide a sparse conditioning structure for message-passing at the finer level. We then leverage equivariant discrete diffusion to model each graph level hierarchically.

Our method enables the efficient generation of large graphs, motivating us to introduce new evaluation datasets with larger graphs and more instances than those in traditional benchmarks. We demonstrate our approach using discrete diffusion, presenting, to the best of our knowledge, the first hierarchical equivariant generative model for graphs.

We summarize our contributions as follows:

- We propose a novel graph coarsening method to produce minimal spanning supergraphs that serve as conditioning and as sparse structures for generation.

- We prove that our coarsening method not only maximizes the sparsity of the conditioned structure but also provides theoretical guarantees for preserving important spectral properties given by the graph spectrum.

- We leverage this method to introduce a hierarchical model based on discrete diffusion, which scales to large graphs with over than a thousand nodes.

- By preserving equivariance in our coarsening and generative framework, we introduce, to the best of our knowledge, the first graph generative approach that is both hierarchical and equivariant.

- Our method inherently provides a conditional generative model for structure-guided graph generation, enabling effective spectrum-conditioned generation.

- We introduce new datasets with more instances and larger graphs, enabling improved evaluation of graph generative models.

Our code is publicly available at https://anonymous.4open.science/r/HD-graph-C557/.

## 2 BACKGROUND

Denoting an unattributed graph as a set of nodes and a set of edges, $\mathcal{G} = (\mathcal{V}_\mathcal{G}, \mathcal{E}_\mathcal{G})$, we begin by introducing two key notions for our method.

**Definition 1.** *$\mathcal{H}$ is a* spanning supergraph *of $\mathcal{G}$, denoted $\mathcal{H} = \mathcal{G} + \mathcal{E}_S$, where $\mathcal{E}_S$ represents a set of additional edges and $+$ denotes the disjoint union. Consequently, we have: $\mathcal{H} = \mathcal{G} + \mathcal{E}_S \iff \mathcal{V}_\mathcal{G} = \mathcal{V}_\mathcal{H}, \mathcal{E}_\mathcal{H} \supseteq \mathcal{E}_\mathcal{G}$.*

**Definition 2.** *The* Gamma Index *of a graph $\mathcal{G}$, denoted $\gamma_\mathcal{G}$, represents the proportion of edges relative to all possible node pairs. For an undirected graph without self-connections, it is defined as: $\gamma = 2m/n(n-1)$.*

We use standard notations for graphs throughout this work. If needed, indications are provided in Appendix A.1.

## 2.1 METHOD OVERVIEW

We present a hierarchical graph generative method that progressively generates finer graphs from coarser ones through a series of conditional generation steps. The conditional generation steps are trained, and operate, on the outputs of a sequence of coarsening and lifting operations. These operations, applied as a preprocessing step, transform an input graph to ever coarser versions of it.

Specifically, for each original input graph in a dataset, we create a sequence of graph representations $\{\mathcal{G}^0, ...\mathcal{G}^L\}$ from the finest level $\mathcal{G}^0$, representing the original input graph to the coarsest level $\mathcal{G}^L$. Additionally, we create a corresponding set of lifted graph representations $\{\mathcal{H}^0, ...\mathcal{H}^{L-1}\}$, where each $\mathcal{H}^l = \text{LIFT}(\mathcal{G}^{l+1}) \neq \mathcal{G}^l$. We highlight two key properties of these hierarchical representations. First, the lifted graph $\mathcal{H}^l$ is a spanning supergraph of its corresponding $\mathcal{G}^l$; and second, unlike the coarsening function, our LIFT function is invertible.

Given that the coarsening process is Markovian and the lifting function is bijective, we model the graph distribution autoregressively over the coarsening levels:

$$p(\mathcal{G}) = p_{\theta_L}(\mathcal{G}^L) \prod_{l=1}^{L} p_{\theta_l}(\mathcal{G}^l|\mathcal{H}^l) = p_{\theta_L}(\mathcal{G}^L) \prod_{l=1}^{L} p_{\theta_l}(\mathcal{G}^l|\mathcal{G}^{l+1}). \tag{1}$$

Figure 1 illustrates one step of our hierarchical approach. Visualizations of preprocessed and generated data are presented in Appendix F.

## 2.2 MOTIVATIONS

Our hierarchical generative model begins at the coarsest level, generating a coarse graph that captures essential high-level spectral properties. The lifted graph, which retains these spectral properties, is then used as a conditioning structure to generate a finer graph enriched with the spectral properties of the lower levels. By iterating this process, we generate progressively finer representations, ultimately capturing the original graph distribution. Instead of generating the entire graph in a single step, our approach incrementally captures both graph structure and spectral information through a sequence of conditional generative steps.

We argue that each of these conditional steps represents a simpler task. Indeed, we leverage the sparsity of the conditioning graph $\mathcal{H}^l$, which serves both as a conditional space and as a message-passing scheme to generate the graph $\mathcal{G}^l$. This sparsity provides several advantages over dense, unconditional models:

1. By leveraging sparsity, our method enables efficient training on larger graphs. As shown by Qin et al. (2024), generative models for graphs require substantial computational power and parameters for edge representations. Since dense models' complexity grows with $\mathcal{O}(n^2)$, large graphs cause memory issues during training, slow training, and slow generation speeds. Moreover, sparsity significantly improves training and generation speed compared to similar dense models, even when applied to relatively small graphs.

2. In contrast to dense models, which predict edge presence or absence for every node pair, our model only needs to predict edges within the conditioning graph $\mathcal{H}$. Consequently, by construction, our approach sets a high proportion $(1 - \gamma_{\mathcal{H}})$ of node pairs as non-edges, preventing the risk of error on those, and focusing the model capacity on the remaining edges.

3. As a result of message-passing, in our model each node aggregates information only from its adjacent nodes in $\mathcal{H}$. In contrast, in dense models, each node aggregates information from every other node in the graph into a fixed-length vector. In large graphs, this process can lead to oversquashing (Topping et al., 2022; Akansha, 2024), limiting their ability to model large graph distributions.

While several hierarchical graph models exist (Jin et al., 2018; Karami, 2024; Bergmeister et al., 2024), none of them preserve equivariance, which, as empirical evidence suggests (Jo et al., 2022), is a crucial property in graph modeling. In contrast, our model preserves equivariance and represents, to the best of our knowledge, the first approach that is both hierarchical and equivariant.

## 2.3 RELATED WORK

We categorize graph generative models into two classes: models that operate sequentially, and equivariant models. Sequential models generate graphs by auto-regressively adding nodes, edges, or subgraphs. (You et al., 2018; Liao et al., 2019; Kong et al., 2023). Equivariant models address the node permutation issue by ensuring a unique computational graph for every possible instantiation of the same object. These models have been developed within various generative frameworks (Martinkus et al., 2022; Liu et al., 2019). Recently, equivariant denoising models (Jo et al., 2022; Haefeli et al., 2022; Vignac et al., 2023; Jo et al., 2024; Eijkelboom et al., 2024) and equivariant quantized autoencoders (Boget et al., 2024; Nguyen et al., 2024) have significantly improved graph generation for small graphs. Similar to the work we present here, SparseDiff (Qin et al., 2024) and EDGE Chen et al. (2023) tackle the scalability challenges faced by equivariant models. A detailed comparison of these frameworks with ours is provided in the Appendix D.1. Recent works (Karami, 2024; Bergmeister et al., 2024) adopt hierarchical approaches. However, none is equivariant. A more comprehensive review of related work is presented in the Appendix.D.1.

## 3 SPANNING SUPERGRAPH GAMMA MINIMIZATION

In this section, we present a new coarsening algorithm and its corresponding lifting function, which we use in a preprocessing step, to create the graph representations $\mathcal{G}$ and $\mathcal{H}$ at different coarseness levels. Our coarsening algorithm brings important advantages in the context of our hierarchical method: It maximizes the sparsity of the spanning supergraph, maintains equivariance, and preserves the graph's spectral properties. We first present relevant graph coarsening work, and motivate the need for a new coarsening method. We then introduce our coarsening algorithm, and derive key properties related to equivariance and spectrum preservation. We validate our method experimentally.

### 3.1 GRAPH COARSENING: RELATED WORKS

Graph coarsening methods can be broadly categorized into three main classes: node-dropping, contraction, and clustering methods. Node-dropping methods (Gao & Ji, 2019; Lee et al., 2019) coarsen a graph by removing specific nodes and their associated edges, either one-by-one or by blocks. Contraction methods (Loukas, 2019; Diehl, 2019) coarsen graphs by merging nodes through either edge contraction, where the endpoints of an edge are merged, or neighborhood contraction, where all adjacent nodes of a given node are merged. However, both methods fail to maintain invariance and equivariance, either during the coarsening process or in their potential reverse lifting scheme (see Appendix C.1 for a detailed discussion).

Clustering methods (Ying et al., 2018; Bianchi et al., 2020) coarsen graphs by merging nodes within the same cluster. They therefore depend on a clustering algorithm. Most clustering-based coarsening methods are permutation-invariant, and they typically focus on community detection-based clustering. In contrast, our approach aims to maximize sparsity. Although our method falls within this category, it differs in its clustering objective. For complete reviews on graph coarsening, we refer to Grattarola et al. (2024); Liu et al. (2023); Wang et al. (2024).

### 3.1.1 COARSENING AND LIFTING GRAPHS

We define coarsening and lifting operations using matrix multiplications. Assume that we have access to an assignment matrix $\boldsymbol{C}_{\mathcal{G}^l} \in \{0,1\}^{n \times K}$, where each row is a one-hot indicator of a node's cluster membership, and $K$ represents the number of clusters.

**Coarsening**  We define $\boldsymbol{n}_{\mathcal{G}}^{l+1} := \boldsymbol{C}_{\mathcal{G}^l}^T \mathbf{1}_n \in \mathbb{N}^K$, the coarse annotation vector. Its elements indicate the number of nodes assigned to each cluster. We denote with $\boldsymbol{N} := \mathrm{diag}(\boldsymbol{n})$ the corresponding diagonal matrix. We define the coarse adjacency matrix as $\boldsymbol{A}_{\mathcal{G}}^{l+1} := \boldsymbol{C}_{\mathcal{G}^l}^{+\,T} \tilde{\boldsymbol{A}}_{\mathcal{G}}^l \boldsymbol{C}_{\mathcal{G}^l}^+ - \boldsymbol{I}_K$, where $\boldsymbol{C}_{\mathcal{G}^l}^+ = (\boldsymbol{C}_{\mathcal{G}^l} \boldsymbol{N}^{-1})^T$ is the left-inverse of $\boldsymbol{C}_{\mathcal{G}^l}$, and $\tilde{\boldsymbol{A}} = \boldsymbol{A} + \boldsymbol{I}_n$ the adjacency matrix augmented with self-connections. Therefore, we define the coarse graph as $\mathcal{G}^{l+1} := (\boldsymbol{n}_{\mathcal{G}}^{l+1}, \boldsymbol{A}_{\mathcal{G}}^{l+1})$.

**Graph lifting** We define the lifted graph as $\mathcal{H}^l := \boldsymbol{A}^l_{\mathcal{H}} = \boldsymbol{C}_{\mathcal{G}^l} \tilde{\boldsymbol{A}}^{l+1}_{\mathcal{G}} \boldsymbol{C}^T_{\mathcal{G}^l} - \boldsymbol{I}_n$. Crucially for our generative model, we observe that we can recover $\boldsymbol{C}_{\mathcal{G}^l}$ up to a permutation uniquely as a function of the coarse annotation vector. We present the procedure in Appendix B.2. Since $\tilde{\boldsymbol{A}}^{l+1}_{\mathcal{G}} = \boldsymbol{C}^+_{\mathcal{G}^l} \boldsymbol{C}_{\mathcal{G}^l} \tilde{\boldsymbol{A}}^{l+1}_{\mathcal{G}} (\boldsymbol{C}^+_{\mathcal{G}^l} \boldsymbol{C}_{\mathcal{G}^l})^T$, we additionally remark that the lift operation is invertible.

For an intuitive and illustrated explanation of the coarsening-lifting process, see Appendix B.1.

**Graph projection** Through the projection matrix $\Pi = \boldsymbol{C}_{\mathcal{G}^l} \boldsymbol{C}^+_{\mathcal{G}^l}$, we can directly obtain the projected graph $\mathcal{H}^l$ a function of $\mathcal{G}^l$: $\mathcal{H} = \boldsymbol{A}^l_{\mathcal{H}} = \Pi \tilde{\boldsymbol{A}}^l_{\mathcal{G}} \Pi^T$. We can now state the following proposition:

**Proposition 1.** *The projected graph $\mathcal{H}$ is a spanning supergraph of $\mathcal{G}$, that is:*

$$\boldsymbol{A}^l_{\mathcal{H}} = \Pi \tilde{\boldsymbol{A}}^l_{\mathcal{G}} \Pi^T \implies \mathcal{H} = \mathcal{G} + \mathcal{E}_{\mathcal{S}}.$$

*Proofs are provided in Appendix A.* □

By coarsening and projecting graphs sequentially, we obtain a dataset with graph representations ordered from finer to coarser $\{(\mathcal{G}^0_i, \mathcal{H}^0_i), ..., (\mathcal{G}^L_i, \mathcal{H}^L_i)\}^N_{i=1}$; at the coarsest level we set $\mathcal{H}^L = \mathcal{K}_{n^L}$, where $\mathcal{K}_n$ is the complete graph (i.e., fully-connected). We set $K$ as a fraction of the number of nodes in the largest graph (between a third and a fifth in our experiments). We stop iterating when the Gamma Index reaches 0.5, which resulted in models with two to four coarseness levels in our experiments. The whole procedure is fast and is performed as a preprocessing step.

## 3.2 Gamma-Min Clustering

Given a graph $\mathcal{G}$, the spanning supergraph is entirely determined by the node clustering, as given by the assignment matrix $\boldsymbol{C}_{\mathcal{G}^l}$. We learn $\boldsymbol{C}_{\mathcal{G}^l}$ by minimizing the Gamma Index of the spanning supergraph. This approach not only maximizes the sparsity of the spanning supergraph, but also ensures equivariance to node permutations, and provides theoretical guarantees for preserving important spectral properties given by the graph spectrum.

To achieve this, we parametrize the assignment matrix $\boldsymbol{C}_{\mathcal{G}^l} \in \{0, 1\}^{n \times K}$ as :

$$\boldsymbol{C}_{\mathcal{G}^l} = \text{Hardmax}(\sigma(\text{GNN}_{\mathcal{G}}(\boldsymbol{X}, \boldsymbol{E}))), \tag{2}$$

where $\sigma$ denotes the softmax function, and Hardmax is the node-wise one-hot encoded argmax, which we implement using the gradient straight-through estimate for backpropagation. The model architecture details are given in the appendix B.

Instead of directly minimizing the number of additional edges in the supergraph, $m_{\mathcal{S}} = |\mathcal{S}|$, we will operate over a normalized version of it using $\frac{m_{\mathcal{S}} + m_{\mathcal{G}}}{m_{\mathcal{K}}} = \frac{m_{\mathcal{H}}}{m_{\mathcal{K}}} = \gamma_{\mathcal{H}}$, which does not depend on the graph size. The normalization yields the Gamma Index of the spanning supergraph. Hence, our objective function, called the Gamma-Min objective, is directly interpretable. Its value is bounded as $\gamma_{\mathcal{G}} \leq \gamma_{\mathcal{H}} \leq 1$. We compute the objective as:

$$\gamma_{\mathcal{H}} = \frac{1}{n(n-1)} \left[ \left( \sum_{i=1}^{K} \sum_{j=1}^{K} W^{l+1}_{i,j} \right) - n \right], \tag{3}$$

where $\boldsymbol{W}^{l+1} = (\boldsymbol{n}^{l+1}_{\mathcal{G}} \boldsymbol{n}^{l+1^T}_{\mathcal{G}}) \odot \tilde{\boldsymbol{A}}^{l+1}_{\mathcal{G}}$, with $\odot$ representing the element-wise product.

## 3.3 Invariance and equivariance

We now state two properties, which are important for our generative model.

**Proposition 2.** *The coarse graph representation $\mathcal{G}^{l+1} = (\boldsymbol{A}^{l+1}_{\mathcal{G}}, \boldsymbol{X}^{l+1}_{\mathcal{G}})$ is invariant to node permutations of the $\mathcal{G}^l$ graph.* □

This invariance, well-known and inherent to clustering-based coarsening methods, ensures that each graph corresponds to a unique coarse graph. In contrast, methods relying on specific orderings, such as edge or node contraction, lack this property and can produce different coarse graphs from the same input graph.

**Proposition 3.** *The spanning supergraph representation $\boldsymbol{A}_{\mathcal{H}}^l$ is equivariant to node permutations of the $\mathcal{G}^l$ graph.* □

This equivariance property ensures that the node representation of a graph and the one of its spanning supergraph are aligned, preventing the need for a costly matching procedure.

### 3.4 SPECTRAL PROPERTIES

We now show that our coarsening-lifting operations preserve important graph features, as given by the graph spectrum. In particular, we prove two properties. First, the lifting operation preserves the spectrum of the graph. Second, minimizing the Gamma Index lower an upper bound on the spectral discrepancy between a graph $\mathcal{G}$ and its projection $\mathcal{H}$. Thus, our method minimizes spectral information loss during coarsening and preserves it entirely during graph lifting.

Formally, we define the spectral distance between a graph and its spanning supergraph as $\sum_{i=1}^{n} |\lambda_{\mathcal{G}}(i) - \lambda_{\mathcal{H}}(i)|$, where $\lambda(i)$ is the $i^{\text{th}}$ eigenvalue of the Laplacian sorted in non-decreasing order.

**Proposition 4.** *The eigenvalues of the normalized Laplacian of the spanning supergraph contain all the eigenvalues of the normalized Laplacian of the weighted coarse graph plus the eigenvalue $1$ with multiplicity $n^l - n^{l+1}$.* □

As a consequence of Proposition 4, the spectral distance between a graph and its spanning supergraph is lower bounded by $\sum_{i=s_1+1}^{s_2} |\lambda_{\mathcal{G}}(i) - 1|$, where $s_1 = \arg\max_i\{i : \lambda_{\mathcal{G}}(i) < 1\}$ and $s_2 = n - K + s_1$.

Let us define the matrix $\boldsymbol{S} := \boldsymbol{A}_{\mathcal{H}}^l - \boldsymbol{A}_{\mathcal{G}}^l$, so that the change of degree of each node between a graph and its spanning supergraph is given by $d_{\mathcal{S}}(i) = \sum_j S_{i,j}$. Then, the following proposition holds.

**Proposition 5.** *The spectral discrepancy between the eigenvalues of the unnormalized Laplacians is upper bounded by twice the maximum degree change that is $|\lambda_{\mathcal{G}}(i) - \lambda_{\mathcal{H}}(i)| \leq 2 \cdot \max_{1 \leq i \leq n} d_{\mathcal{S}}(i)$.* □

By minimizing $\gamma_{\mathcal{H}}$, we minimize an upper bound of the spectral discrepancy between the input graph and its spanning supergraph. Since lifting the graph preserves the spectrum, we interpret the discrepancy as an indication of the information loss resulting from the graph's coarsening. So by minimizing $\gamma_{\mathcal{H}}$, we minimize this loss.

### 3.5 EMPIRICAL EVALUATION

Our method not only provides theoretical guarantees in terms of structural information preservation but also demonstrates empirical efficiency. Table 1 reports the average Gamma Index of the spanning supergraphs obtained after the first coarsening step across various datasets, comparing our model (GammaMin) with two popular deep coarsening methods: DiffPool (Ying et al., 2018) and MinCut (Bianchi et al., 2020). Detailed descriptions of the datasets and experimental settings, including baseline descriptions, are provided in Appendices E.4, and B.3, respectively.

As our model directly minimizes the Gamma Index, it is expected to perform better on this objective. However, these empirical results highlight the importance of developing a coarsening method specifically designed for our hierarchical generative approach.

Table 1: Gamma Index for three coarsening models.

| Dataset | Data | DiffPool | MinCut | $\gamma$-Min (Ours) |
|---|---|---|---|---|
| Zinc250k | 0.094 | 0.336 | 0.305 | 0.247 |
| SBM20k | 0.083 | 0.640 | 0.265 | 0.213 |
| GitStar | 0.016 | 0.955 | 0.249 | 0.053 |
| Reddit12k | 0.005 | 0.857 | 0.067 | 0.017 |

The Gamma Index also represents the ratio of edges in our model compared to dense models that operate on the fully connected graphs. Our method significantly reduces this ratio, which directly corresponds to a reduction in the computational complexity of the model. This advantage is especially pronounced for large, sparse graphs. The resulting sparsity enables our model to efficiently scale in training and inference, handling graphs with more than a thousand nodes.

## 4 Hierarchical Equivariant Discrete Diffusion

We introduce our $\underline{H}$ierarchical $\underline{E}$quivariant $\underline{D}$iscrete $\underline{D}$iffusion method (HEDD), which models the graph distribution as an auto-regressive sequence of graphs, as expressed in Equation 1. After pre-processing, the dataset contains graph pairs for each level: $\{(\mathcal{G}_i^0, \mathcal{H}_i^0), ..., (\mathcal{G}_i^L, \mathcal{H}_i^L)\}_{i=1}^N$. At each level $l$, we independently model the conditional distribution $p(\mathcal{G}^l|\mathcal{H}^l)$. While we use discrete diffusion to model these conditional distributions, our hierarchical framework is compatible with other equivariant generative approaches.

### 4.1 Background

Discrete diffusion (Austin et al., 2021) consists of a forward process that progressively adds discrete noise to an instance until it reaches a known limit distribution and a backward process where a model is trained to iteratively denoise the instances. In our method, both processes operate at each coarseness level independently. At the coarsest level, $\mathcal{H}^L$ is set as $\mathcal{K}_{n^L}$, making the conditional formulation equivalent to the unconditional one[1]: $p_{\theta^L}(\mathcal{G}|\mathcal{H} = \mathcal{K}_{n^L}) = p_{\theta^L}(\mathcal{G})$. All levels sharing the same formulation, we simplify notation by omitting the superscript indicating the level.

The key idea is to create an edge attribute matrix $\boldsymbol{E} \in \mathbb{R}^{m \times d_e}$, representing the edge attributes of $\mathcal{H}$, to encode the edge attributes of $\mathcal{G}$. We make this possible thanks to the fact that $\mathcal{H}$ is a spanning supergraph $\mathcal{G}$ (Proposition 1), that $\mathcal{H}$ is equivariant to $\mathcal{G}$ (Proposition 3), and that $\mathcal{H}$ is an unattributed graph. To encode $\mathcal{G}$ into $\boldsymbol{E}$, we treat the absence of an edge as a distinct edge type. Specifically, each row vector $E_i$ is a one-hot encoding of the edge attribute in $\mathcal{G}$, with one category representing the absence of an edge. The correspondence between edges and their attributes is maintained via the edge index matrix $\boldsymbol{J} \in \mathbb{N}^{2 \times m}$. Similarly, thanks to the fact that $\mathcal{H}$ is unattributed, the annotation matrix $\boldsymbol{X}$ represents the node attributes of $\mathcal{G}$. Since both graphs share the same set of nodes $\mathcal{V}_\mathcal{G} = \mathcal{E}_\mathcal{H}$, there is no need to encode the absence of a node.

### 4.2 Sparse Noising Process

We define edge and node type transition probabilities between time steps $t-1$ and $t$ via the transition matrices $\boldsymbol{Q}_{\boldsymbol{E}}^t \in \mathbb{R}^{d_e \times d_e}$, and $\boldsymbol{Q}_{\boldsymbol{X}}^t \in \mathbb{R}^{d_x \times d_x}$, respectively. The noising process consists of sampling each node and each edge independently from the following categorical distributions:

$$q(\boldsymbol{X}^t|\boldsymbol{X}^0) = \boldsymbol{X}^0 \bar{\boldsymbol{Q}}_X^t, \text{and} \quad q(\boldsymbol{E}^t|\boldsymbol{E}^0) = \boldsymbol{E}^0 \bar{\boldsymbol{Q}}_E^t, \tag{4}$$

where $\bar{\boldsymbol{Q}}_.^t = \prod_{t=0}^{t-1} \boldsymbol{Q}_{..}^t$.

### 4.3 Denoising

Since $\boldsymbol{X}^t$, $\boldsymbol{E}^t$, and $\boldsymbol{J}_\mathcal{H}$ jointly represent $\mathcal{G}^t$ and $\mathcal{H}$, we leverage the sparsity in $\mathcal{H}$ to parametrize $p_\theta(\mathcal{G}^0|\mathcal{G}^t, \mathcal{H})$ using a Message-Passing Neural Networks (MPNN). Our model takes as input the noisy annotation and edge attribute matrices $\boldsymbol{X}^t$ and $\boldsymbol{E}_\mathcal{H}^t$, the time step $t$ as well as the edge index matrix $\boldsymbol{J}_\mathcal{H}$, which indicates the edges involved in the message-passing scheme. It outputs the predicted annotation and edge attribute matrices $\tilde{\boldsymbol{X}}$ and $\tilde{\boldsymbol{E}}$: $\tilde{\boldsymbol{X}}, \tilde{\boldsymbol{E}} = \sigma(\text{MPNN}_\mathcal{H}(\boldsymbol{X}^t, \boldsymbol{E}_\mathcal{H}^t, t; \boldsymbol{J}_\mathcal{H}))$, where $\sigma$ is the sigmoid function for binary variables and the softmax function otherwise. We interpret each model output as probabilities over the clean graph $\tilde{E}_i = p_\theta(E_i|\mathcal{H}, \mathcal{G}^t)$ and $\tilde{X}_i = p_\theta(X_i|\mathcal{H}, \mathcal{G}^t)$. We train our model by maximizing its log-likelihood, which coincide with the cross-entropy loss used in Vignac et al. (2023):

$$\mathcal{L} = \mathbb{E}_{(\mathcal{G}, \mathcal{H}) \sim p_{\text{data}}} \left[ \mathbb{E}_{q(X_i^t|X_i^0)} \gamma[-\log(p_\theta(X_i|\mathcal{H}, \mathcal{G}^t))] + \mathbb{E}_{q(E_i^t|E_i^0)}(1 - \gamma)[-\log(p_\theta(E_j|\mathcal{H}, \mathcal{G}^t))] \right],$$
$$\tag{5}$$

where $\gamma$ is a weighting factor between nodes and edges. We use $\gamma = n/(n + m)$.

---

[1]Unconditional discrete diffusion models are actually implicitly conditioned on the number of nodes n.

## 4.4 SAMPLING

We first describe how we sample on a single level, and then describe the hierarchical sampling process.

**Level-wise sampling**   To simplify the presentation, let us first assume access to $\mathcal{H}$. Sampling at a single level requires estimating the reverse diffusion $p(\mathcal{G}^{t-1}|\mathcal{G}^t, \mathcal{H})$, which we model as a product over nodes and edges (see Equation 41). This probability distribution is approximated by marginalizing over the network prediction, as in standard discrete diffusion, following Equation 42.

In practice, we first sample $\boldsymbol{E}^T$ and $\boldsymbol{X}^T$ from the known limit distributions, according to the edges and nodes induced by $\boldsymbol{J}_{\mathcal{H}}$. We then denoise the graph iteratively using Equation 42 until we obtain $\hat{\boldsymbol{E}}^0$ and $\hat{\boldsymbol{X}}^0$, from which $\hat{\mathcal{G}}$ is constructed. During sampling, however, $\mathcal{H}$ is not available. Instead, following a teacher-forcing strategy, we replace it with the $\hat{\mathcal{H}}$, induced by the graph sampled at the previous coarser level.

**Hierarchical sampling**   At the coarsest level $L$, we sample $n^L$ according to the data distribution and set $\mathcal{H}^L = \mathcal{K}_{n^L}$. Using our model for level $L$, we sample $\hat{\mathcal{G}}^L$. Next, we lift $\hat{\mathcal{G}}^L$ to obtain $\hat{\mathcal{H}}^{L-1}$ using $\tilde{\boldsymbol{A}}_{\hat{\mathcal{H}}}^{L-1} = \hat{\boldsymbol{C}}_{\hat{\mathcal{G}}}^L \tilde{\boldsymbol{A}}_{\hat{\mathcal{G}}}^L (\hat{\boldsymbol{C}}_{\hat{\mathcal{G}}}^L)^T$, where $\hat{\boldsymbol{C}}_{\hat{\mathcal{G}}}^L$ is the assignment matrix reconstructed from $\hat{\mathcal{G}}^L$ (this procedure is detailed in Appendix B.2). Using $\hat{\mathcal{H}}^{L-1}$, which is fully described by $\tilde{\boldsymbol{A}}_{\hat{\mathcal{H}}}^{L-1}$, we sample $\hat{\mathcal{G}}^{L-1}$ from level $L-1$. By iterating this procedure across all levels, we eventually generate $\hat{\mathcal{G}} = \hat{\mathcal{G}}^0$. For node- and edge-attributed graphs, our model outputs their attributes generating the finest coarseness level.

## 4.5 PROPERTIES

**Complexity**   By conditioning on $\mathcal{H}$, we reduce the computational complexity from $\mathcal{O}(n^2)$ (standard discrete diffusion) to $\mathcal{O}(m_{\mathcal{H}})$, linear in the number of edges of $\mathcal{H}$.

**Equivariance**   Since we apply the noise independently to each node and edge, and the conditioning spanning supergraph is equivariant to node permutations, our denoising model inherits this equivariance.

**Additional Graph Features**   Enhancing GNN inputs by computing additional synthetic features, including spectral embeddings, has become a widespread practice (Martinkus et al., 2022; Vignac et al., 2023; Qin et al., 2024; Bergmeister et al., 2024; Boget et al., 2024). However, denoising models usually need to recompute these features before each forward pass, both during training generation, which is computationally expensive. Instead, we use the spectral embeddings of the spanning supergraph, the graph size and the cluster size as extra feature, which we need to compute only once during preprocessing. We present the detailed extra features scheme in appendix B.4.

## 4.6 CONDITIONAL GENERATION

Conditional generation is an important feature for generative models, as most real-world applications require some form of guidance during the generation process. Notably, our model inherently operates as a structure-conditioning model. By fixing the conditioning graph $\mathcal{H}$, our model restricts the generated graphs to those satisfying $\mathcal{G} = \mathcal{H} \setminus \mathcal{E}_S$. We are not aware of models proposing a similar structural conditioning. We demonstrate in Section 5 the effectiveness of our conditional generative model in guiding the graph spectrum during generation.

A more common conditional generation setting consists of generating molecules with specific global properties, represented as a vector $\boldsymbol{c}$. By concatenating this conditioning vector to each node attribute input vector, we obtain a property-based conditional generative model. Furthermore, by training both conditional and unconditional models, we would enable classifier-free guidance (Ho & Salimans, 2022). However, this feature has not yet been implemented in our current work.

## 5 EXPERIMENTS

We evaluate our model on molecular, synthetic, and real graph datasets. For molecules, we use the Zinc250k dataset, containing 250,000 molecular graphs with up to 38 heavy atoms of 9 types.

For large graphs, we introduce three new datasets that not only contain larger graphs but also have large numbers of instances: the Stochastic Block Model 20k (SBM20k), Github Stargazers (GitStar), and Reddit 12k, which contain 20,000, 12,725, and 11,551 graphs with up to 194, 957, and 1,499 nodes, respectively. Indeed, most popular benchmark datasets for large graphs, i.e., the Stochastic Block Model (SBM), Ego, and Proteins, contain few instances (200, 720, 916, respectively), which raises two main issues. First, any models with enough capacity can overfit the datasets, producing good evaluation results independently of their generalization ability. Second, by saving a small proportion of the data as a test set, the evaluation relies on few instances, making it untrustworthy.

We ran all experiments on a 25GB GPU. Time indications are clock time in seconds. Due to space limitation, we provide detailed descriptions of both the evaluation procedure and the datasets in Appendix E.

### 5.1 MOLECULE GENERATION

For molecule generation, we compare our model with four recent competitive models: DGAE (Boget et al., 2024), a discrete equivariant auto-encoder, GDSS (Jo et al., 2022), a continuous score-based model, DiGress (Vignac et al., 2023), and SparseDiff (Qin et al., 2024), which are discrete diffusion models. For ablation, we also implement a dense discrete diffusion (DiscDiff) model using exactly the same architecture and parameterization as the finer level of our HEDD. We report the Fréchet ChemNet Distance (FCD) (Preuer et al., 2018), which evaluates the similarity of the generated molecules to real molecules in chemical space, and the Neighborhood subgraph pairwise distance kernel (NSPDK) (Costa & Grave, 2010) metrics, comparing their graph structures. Validity without correction indicates the proportion of chemically valid molecules. We report the less informative uniqueness, and novelty in the Appendix E.

The results, reported in Table 2, are the means and standard deviations of three runs. For DGAE and GDSS, we used the results from the original article. The results show that our method better captures the chemical and graph structures than other models while maintaining a high validity rate. We attribute the chemical and graph structure improvement to the structure conditioning brought by our method. Despite the relatively small graph size in Zinc250k, we also observe that our method significantly speeds up graph generation compared to other discrete diffusion models.

Table 2: Generation results on the **Zinc** dataset. The NSPDK results are rescaled by $10^3$.

| Model | NSPDK ↓ | FCD↓ | Val. wo. corr. %↑ | Gen. Time (s) ↓ |
|---|---|---|---|---|
| DGAE | $7 \pm 0$ | $4.4 \pm 0.0$ | $77.9 \pm 0.5$ | - |
| GDSS | $19 \pm 1$ | $14.7 \pm 0.7$ | $97.0 \pm 0.8$ | - |
| DiGress | $71.3 \pm 1.3$ | $18.80 \pm 0.19$ | $90.75 \pm 0.51$ | $5517 \pm 29$ |
| SparseDiff | $55.1 \pm 1.8$ | $15.82 \pm 0.16$ | $76.05 \pm 1.47$ | $9826 \pm 85$ |
| DiscDiff | $11.4 \pm 0.5$ | $7.36 \pm 0.09$ | $\mathbf{97.74} \pm 0.80$ | $4372 \pm 40$ |
| HEDD | $\mathbf{2.1} \pm 0.2$ | $\mathbf{3.80} \pm 0.07$ | $97.07 \pm 0.22$ | $\mathbf{465} \pm 5$ |

### 5.2 LARGE GRAPH GENERATION

For large unannotated graph datasets (SBM20K, GitStar, Reddit12K), we compare our HEDD with dense equivariant models (DiGress, DiscDiff), and recent models designed to scale to large graphs SparseDiff, EDGE, (Chen et al., 2023) and GraphLE (Bergmeister et al., 2024). Unfortunately, we were unable to generate graphs in a reasonable amount of time with GraphLE. As expected, dense models cannot train on larger graphs, resulting in Out-Of-Memory (OOM) errors.

We report the usual Maximum Mean Discrepancies (MMDs) computed over the degree, clustering (cluster.), orbit, and spectrum (spect.) distributions, as well as the generation time (gen. t.) and training time per epoch (ep. t.). For SBM20K, we computed MMDs and generation times over 1000 generated graphs and 1000 graphs from the test set. Due to the slow generation time from SparseDiff, we used only 100 graphs for evaluation on the GitStar and Reddit12K datasets. Still, we

Table 3: Generation results on large graph datasets. The MMDs results are rescaled by $10^3$.

| | Model | degree↓ | clust. ↓ | orbit↓ | spect.↓ | gen. t. (s)↓ | ep. t. (s)↓ |
|---|---|---|---|---|---|---|---|
| **SBM20K** | EDGE | 154.16 ± 3.79 | 766.79 ± 19.24 | **10.58** ± 0.12 | 37.63 ± 1.63 | **350** ± 8 | 430 ± 2 |
| | DiGress | 4.85 ± 6.15 | 5.92 ± 1.97 | 28.30 ± 14.4 | 1.30 ± 1.15 | 3906 ± 53 | 626 ± 1 |
| | SparseDiff | 1.10 ± 0.34 | 5.25 ± 0.50 | 19.11 ± 4.52 | **0.54** ± 0.12 | 11089 ± 640 | 241 ± 0 |
| | DiscDiff | 1.68 ± 0.72 | 23.37 ± 0.44 | 74.65 ± 0.99 | 5.44 ± 0.48 | 3594 ± 8 | 267 ± 0 |
| | HEDD | **0.18** ± 0.02 | **4.98** ± 0.38 | 16.26 ± 3.90 | 1.31 ± 0.34 | 526 ± 18 | **70** ± 0 |
| **GitStar** | EDGE | 32.58 ± 7.18 | 60.39 ± 2.97 | **11.30** ± 0.60 | 12.61 ± 0.70 | 103.7 ± 1.9 | 249 ± 3 |
| | DiGress | OOM | - | - | - | - | - |
| | SparseDiff | 11.52 ± 6.05 | 32.92 ± 6.37 | 25.79 ± 4.47 | 17.36 ± 6.84 | 2715.0 ± 205.3 | 1119 ± 6 |
| | DiscDiff | OOM | - | - | - | - | - |
| | HEDD | **0.52** ± 0.15 | **15.20** ± 0.23 | 11.51 ± 1.60 | **2.42** ± 0.60 | **52.0** ± 15.5 | **56** ± 1 |
| **Reddit** | EDGE | 154.16 ± 3.79 | 766.79 ± 19.24 | **10.58** ± 0.12 | 37.63 ± 1.63 | **172** ± 1 | 606 ± 5 |
| | SparseDiff | 93.73 ± 10.80 | 68.46 ± 39.01 | 202.48 ± 10.75 | 124.96 ± 14.24 | 7234 ± 859 | 1350 ± 18 |
| | HEDD | **3.50** ± 2.54 | **7.33** ± 1.53 | 41.94 ± 11.46 | **6.27** ± 2.27 | 177 ± 9 | **93** ± 0 |

provide results for our method with 1000 graphs in the Appendix E for future comparison. We report the results in Tables 3. For our hierarchical model, generation and epoch times are the cumulated time across all levels.

We observe that our method better captures the graph distribution. On the SBM20k dataset, SparseDiff and DiGress produce results comparable to our HEDD but at the cost of much slower generation. For the datasets with the largest graphs (GitStar, Reddit), the benefits of our method are even more apparent, both in terms of convergence and generation time. Visualizations are available in Appendix F.

### 5.3 ABLATION AND CONDITIONAL GENERATION

**Number of levels** From Tables 2 and 3, we observe that on the Zinc and SBM20k datasets, our hierarchical model with 2 and 3 levels, respectively, consistently outperforms their non-hierarchical counterparts (DiscDiff), which represents a 1-level model. On larger datasets, a minimum of 3 levels was required to avoid out-of-memory issues. In Appendix E.2, we provide a complete ablation study on the number of levels. We observe a strong relation between the sparsity of the conditioning graph and the number of levels. However, no clear or systematic relationship in generative performance was observed.

**Conditional generation** Experimental results demonstrate the effectiveness of our model in conditioning on graph structure. The average spectral distance between reference graphs ($\mathcal{G}_{\text{ref}}$) and conditionally generated graphs ($\hat{\mathcal{G}} \sim p_\theta(\mathcal{G}|\mathcal{H})$) is nearly ten times smaller than that between reference graphs and test set graphs (see detailed experimental settings and results in Appendix E.3).

## 6 CONCLUSION

We demonstrate that our method, leveraging discrete diffusion, enables fast and scalable graph generation. Our approach more effectively captures large graph distributions by extracting global structural features and learning them hierarchically. In addition, it generates graphs significantly faster than other methods and reduces training time. At this stage, we do not foresee any significant ethical concerns related to our model.

We acknowledge that our methods requires training on multiple graph levels, which, despite the method's robustness to hyperparameter changes, expands the hyperparameter space and makes fine-tuning potentially challenging. However, we emphasize that this multi-level approach offers several benefits, including: accelerating the overall training and generation process, enabling generation on large graphs, and significantly improving generative performance compared to the few available baseline methods.

Orthogonal to improvements regarding the generative framework itself, our hierarchical method highlights the importance of model architecture in graph generation. Hierarchical equivariant models hold great potential for advancing graph generation and scaling it to even larger graphs.

## REPRODUCIBILITY STATEMENT

Reproducibility, as a cornerstone of science, is essential to us. Below, we outline the efforts made to facilitate the reproduction of our results:

- Code Availability: Our code is publicly available at https://anonymous.4open.science/r/HD-graph-C557/. It includes the complete model code, dataset preprocessing scripts, and default configuration files for all experiments, along with installation instructions.

- Model Details: The full model architecture, hyperparameters, and the extra feature scheme are provided in Section B of the Appendix.

- Dataset and Evaluation: We describe the datasets and detailed evaluation procedures, along with references, in Section E of the Appendix. Links to the files containing the exact training and test set splits used in our experiments are also provided.

- New Dataset: We have made the new synthetic dataset, Stochastic Block Model 20k, publicly available. Details on how it was created can be found in Section E of the Appendix, and the code to produce it is included in our repository.

- Benchmark Models: We exclusively used benchmark models with official public repositories. The hyperparameters used for our experiments are detailed in Section E of the Appendix.

- Proofs: All proofs of the propositions presented in the paper are included in Section A of the Appendix.

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

# A PROOFS

## A.1 PRELIMINARIES

### A.1.1 NOTATION

We define a graph as $\mathcal{G} = (\mathcal{V}, \mathcal{E}, X, E)$, where $\mathcal{V}$ is a set of nodes, $\mathcal{E}$ is a set of edges, $X$ and $E$ are functions mapping nodes and edges to their respective features. We denote the number of nodes and the number of edges as $n = |\mathcal{V}|$ and $m = \mathcal{E}$, respectively. An unattributed graph can be represented in one of two ways: as an adjacency matrix $\boldsymbol{A} \in \{0,1\}^{n \times n}$, or as a matrix of edge indices $\boldsymbol{J} \in \mathbb{N}^{2 \times m}$, where the first row contains the indices of source nodes and the second the indices of the corresponding target nodes. Edge attributes, of dimension $d_e$ are represented as a matrix of edge attributes, $\boldsymbol{E} \in \mathbb{R}^{m \times d_e}$, where the attributes $E_{i,\cdot}$ correspond to edge $J_{\cdot,i}$. Node attributes, of dimension $d_x$ are represented as an annotation matrix $\boldsymbol{X} \in \mathbb{R}^{n \times d_x}$. We denote $\tilde{\boldsymbol{A}} = \boldsymbol{A} + \boldsymbol{I}_n$ the adjacency matrix with virtual additional self-connections.

Let $\mathcal{G} = (\mathcal{V}, \mathcal{E}, X, E)$ be an undirected graph, where $X$ and $E$ are functions mapping nodes and edges to their respective feature vectors such that $X(\nu_i) = \boldsymbol{x}_i \in \mathbb{R}^{d_x}$ and $E(\nu_i, \nu_j) = \boldsymbol{e}_{i,j} \in \mathbb{R}^{d_e}$. We can also represent a graph with the triplet of its adjacency matrix, $\boldsymbol{A} \in \{0,1\}^{n \times n}$, its annotation matrix, $\boldsymbol{X} \in \mathbb{R}^{n \times d_x}$, and its matrix of edge attributes, $\boldsymbol{E} \in \mathbb{R}^{m \times d_e}$, where $n$ and $m$ are the number of nodes and edges respectively, and $d_x$ and $d_e$ are the dimensions of the node and edge attribute vectors. Denoting $+$ the union of disjoint subsets, we remind that $\mathcal{H}$ is a spanning supergraph of $G$ if $\mathcal{H} = \mathcal{G} + \mathcal{S}$, where $\mathcal{S}$ is a set of additional edges. We denote by $\mathcal{K}$ the fully connected graph, i.e. the graph such that $\mathcal{E}_{\mathcal{K}} = \{(\nu_i, \nu_j) | \forall\ \nu_i, \nu_j \in \mathcal{V}_{\mathcal{K}}\}$.

### A.1.2 COARSENING-LIFTING

For the subsequent proofs, we first formalize the coarsening and lifting procedure in terms of node and edge sets.

A node partitioning of a $\mathcal{G}$ graph in $K$ non-overlapping subsets of vertices such that $\bigcup_{k=1}^{K} \mathcal{C}_k = \mathcal{V}$ and $\bigcap_{k=1}^{K} \mathcal{C}_k = \emptyset$ results from an assignment function $C$ mapping each node to a partition $C(\nu_j) = \mathcal{C}_i$. We can represent the node partitioning as an assignment matrix $\boldsymbol{C} \in \{0,1\}^{n \times K}$, where the row vector corresponds to one-hot encoded node assignments. The node partitioning also structures the edges to subsets of edges that link the nodes of some partition $k$ to the notes of another partition $l$, i.e. $\mathcal{B}_{k,l} = \{(\nu_i, \nu_j) | \nu_i \in \mathcal{C}_k, \nu_j \in \mathcal{C}_l, (\nu_i, \nu_j) \in \mathcal{E}\}$.

Given a node partitioning of a $\mathcal{G}^l$ graph we can define a graph coarsening operation, $f(\mathcal{G}^l)$, which will produce a coarse version, $\mathcal{G}^{l+1}$, of the original graph in which each partition of the original graph gives rise to a node in the coarse graph and the nodes of the coarse graph are connected if there is an edge between the nodes contained in their respective partitions. More formally, $f(\mathcal{G}^l) = \mathcal{G}^{l+1} = (\mathcal{V}^{l+1}, \mathcal{E}^{l+1}, X^{l+1})$ such that $\nu_i^{l+1} \in \mathcal{V}^{l+1} \iff \mathcal{C}_i^l \neq \emptyset, (\nu_i^{l+1}, \nu_j^{l+1}) \in \mathcal{E}^{l+1} \iff \mathcal{B}_{i,j}^l \neq \emptyset$ and, where $X$ is the annotation function $X(\nu_i^{l+1}) = x_i^{l+1} = |\mathcal{C}_i^l| = n_i^l$. So, the vector of partition cardinalities $\boldsymbol{n} = [n_1, ..., n_K]^T \in \mathbb{N}^K$ is also the one-dimensional annotation matrix.

The lifting function, $g(.)$, operates in the opposite direction; given a coarse graph $\mathcal{G}^{l+1}$, it produces a finer one, $\mathcal{H}^l$, where each node of the coarse graph gives rise to a clique, whose node cardinality is provided by the annotation of the node, and pairs of connected nodes in the coarse graph produce bicliques. More formally, $g(\mathcal{G}^{l+1}) = \mathcal{H}^l = (\mathcal{V}_{\mathcal{H}}^l, \mathcal{E}_{\mathcal{H}}^l)$, such that $\mathcal{V}_{\mathcal{H}}^l = \{\mathcal{C}_1, ...\mathcal{C}_C\}$, where $\mathcal{C}_i = \{\nu_j | j \in \mathbb{N}, (\sum_{r=1}^{i} n_{r-1}) + 1 \leq j \leq \sum_{r=1}^{i} n_r, n_0 = 0\}$ and $\mathcal{E}_{\mathcal{H}}^l = \{(\nu_i, \nu_j) | \nu_i, \nu_j \in \mathcal{C}_k\} \cup \{(\nu_i, \nu_j) | \nu_i \in \mathcal{C}_s, \nu_j \in \mathcal{C}_r, (\nu_s, \nu_r) \in \mathcal{E}_{\mathcal{G}}^{l+1}\}$.

We will denote by $h$ the composition of the coarsening and lifting function $h = f \circ g$. Given the $\mathcal{G}^l$ graph, $h(\mathcal{G}^l)$ produces the lifted graph $\mathcal{H}^l$ which has the same nodes as $\mathcal{G}^l$ and its edge set is a superset of those of $\mathcal{G}^l$; as we will see right away $h(\mathcal{G}^l)$ is a spanning supergraph of $\mathcal{G}^l$.

## A.2 PROPOSITION 1

To prove that $\mathcal{H} = \mathcal{G} + \mathcal{S}$, it is sufficient to show that $\mathcal{V}_{\mathcal{H}} = \mathcal{V}_{\mathcal{G}}$ and $\mathcal{E}_{\mathcal{H}} \supseteq \mathcal{E}_{\mathcal{G}}$.

**1. We show: $\mathcal{V}_\mathcal{H} = \mathcal{V}_\mathcal{G}$.**

Let call $\mathcal{C}^C = \{\mathcal{C}_1^C, ..., \mathcal{C}_K^C\}$ and $\mathcal{C}^L = \{\mathcal{C}_1^L, ..., \mathcal{C}_K^L\}$ the clustering resulting from coarsening and lifting, respectively.

We have $\mathcal{V}_\mathcal{G} = \{\nu_i | \nu_i \in C_k^C, \forall k \in [K]\}$ and $\mathcal{V}_\mathcal{H} = \{\nu_i | \nu_i \in C_k^L, \forall k \in [K]\}$.

Let call $\pi$ the node permutation such that:
$\pi(\mathcal{C}_i^C) = \{\nu_{\pi(j)} | \pi(j) \in \mathbb{N}, (\sum_{r=1}^i n_{r-1}) + 1 \le \pi(j) \le \sum_{r=1}^i n_r, n_0 = 0\}$

Then, we have: $\pi(\mathcal{C}_i^C) = \mathcal{C}_i^L, \quad \forall i \in [K]$.

Hence, $\mathcal{V}_\mathcal{H} = \mathcal{V}_\mathcal{G}$.

**2. We show: $\mathcal{E}_\mathcal{H} \supseteq \mathcal{E}_\mathcal{G}$.**

Let assume $(\nu_i, \nu_j) \in \mathcal{E}_\mathcal{G}$

Case 1: $\{\nu_i, \nu_j\} \subseteq \mathcal{C}_k^C$

$$\{\nu_i, \nu_j\} \subseteq \mathcal{C}_k^C \iff \{\nu_{\pi(i)}, \nu_{\pi(j)}\} \subseteq \pi(\mathcal{C}_k^C) \tag{6}$$
$$\iff \{\nu_i, \nu_j\} \subseteq \mathcal{C}_k^L \tag{7}$$
$$\implies (\nu_i, \nu_j) \in \mathcal{E}_\mathcal{H} \tag{8}$$

Case 2: $\nu_i \in \mathcal{C}_a^C, \nu_j \in \mathcal{C}_b^C$

$$\nu_i \in \mathcal{C}_a^C, \nu_j \in \mathcal{C}_b^C \iff \nu_{\pi(i)} \in \pi(\mathcal{C}_a^C), \nu_{\pi(j)} \in \pi(\mathcal{C}_b^C) \tag{9}$$
$$\iff \nu_i \in \mathcal{C}_a^L, \nu_j \in \mathcal{C}_b^L \tag{10}$$
$$\nu_{\pi(i)} \in \pi(\mathcal{C}_a^C), \nu_{\pi(j)} \in \pi(\mathcal{C}_b^C) \implies \mathcal{B}_{a,b} \ne \emptyset \tag{11}$$
$$\iff (\nu_a, \nu_b) \in \mathcal{E}_\mathcal{G}^{l+1} \tag{12}$$
$$\nu_i \in \mathcal{C}_a^L, \nu_j \in \mathcal{C}_b^L, (\nu_a, \nu_b) \in \mathcal{E}_\mathcal{G}^{l+1} \implies (\nu_i, \nu_j) \in \mathcal{E}_\mathcal{H} \tag{13}$$

In both cases: $(\nu_i, \nu_j) \in \mathcal{E}_\mathcal{G} \implies (\nu_i, \nu_j) \in \mathcal{E}_\mathcal{H}$

Hence, $\mathcal{E}_\mathcal{G} \subseteq \mathcal{E}_\mathcal{H}$ $\qquad\qquad\qquad\square$

### A.3 PROPOSITION 2

PRELIMINARIES

Without loss of generality, we represent the permutation $\pi \in \Pi$, with $\Pi$ being the set of all possible permutations, by the matrix $\boldsymbol{P}_\pi$.

Note that $\boldsymbol{C}$ is the composition of an MPNN, which is equivariant by construction, and element-wise operations that are trivially equivariant. Hence, we have:
$$\boldsymbol{P}_\pi \boldsymbol{C} = \text{Hardmax}(\sigma(\text{GNN}_\mathcal{G}(\boldsymbol{P}_\pi \boldsymbol{X}, \boldsymbol{E}_\pi))) \quad \forall \pi \in \Pi. \tag{14}$$

Note that $\boldsymbol{N} = diag(\boldsymbol{n}) = diag(\boldsymbol{C}^T \boldsymbol{1})$ is invariant to node permutation, because:
$$\boldsymbol{C}^T \boldsymbol{1} = \boldsymbol{n} \implies \boldsymbol{C}^T \boldsymbol{P}_\pi^T \boldsymbol{1} = \boldsymbol{n} \tag{15}$$

PROOF

We show that the coarse graph representation $\mathcal{G}^{l+1} = (\boldsymbol{A}_\mathcal{G}^{l+1}, \boldsymbol{X}_\mathcal{G}^{l+1})$ is invariant to node permutation of $\mathcal{G}^l = (\boldsymbol{A}_\mathcal{G}^l, \boldsymbol{X}_\mathcal{G}^1)$

Let define:
$$\boldsymbol{A}_\mathcal{G}^{l+1} = f_A(\boldsymbol{A}_\mathcal{G}^l) = (\boldsymbol{C}\boldsymbol{N}^{-1})^T \boldsymbol{A}_\mathcal{G}^l (\boldsymbol{C}\boldsymbol{N}^{-1}) \tag{16}$$
$$\boldsymbol{X}_\mathcal{G}^{l+1} = f_X(\boldsymbol{C}) = \boldsymbol{C}^T \boldsymbol{1} \tag{17}$$

We show that:

$$A_{\mathcal{G}}^{l+1} = f_A(A_{\mathcal{G}}^l) \implies A_{\mathcal{G}}^{l+1} = f_A(P_\pi A_{\mathcal{G}}^l P_\pi^T) \tag{18}$$

$$X_{\mathcal{G}}^{l+1} = f_X(A_{\mathcal{G}}^l) \implies X_{\mathcal{G}}^{l+1} = f_X(P_\pi A_{\mathcal{G}}^l P_\pi^T) \tag{19}$$

$$f_A(P_\pi^T A_{\mathcal{G}}^l P_\pi) = (P_\pi C N^{-1})^T P_\pi A_{\mathcal{G}}^l P_\pi^T (P_\pi C N^{-1}) \tag{20}$$

$$= (C N^{-1})^T P_\pi^T P_\pi A_{\mathcal{G}}^l P_\pi^T P_\pi (C N^{-1}) \tag{21}$$

$$= (C N^{-1})^T A_{\mathcal{G}}^l (C N^{-1}) \tag{22}$$

$$= f_A(A_{\mathcal{G}}^l) \tag{23}$$

Equation 22 uses the fact that permutation matrices are orthonormal matrices, such that $P_\pi^T P_\pi = P_\pi P_\pi^T = I$.

The invariance of $f_X$ follows directly from Equation 15. □

## A.4 PROPOSITION 3

Let define $A_{\mathcal{H}}^l = h_A(A_{\mathcal{G}}^l) = C A_{\mathcal{G}}^{l+1} C^T$

We show that:

$$h_A(P_\pi A_{\mathcal{G}}^l P_\pi^T) = .P_\pi A_{\mathcal{H}}^l P_\pi^T \quad \forall \pi \in \Pi \tag{24}$$

$$h_A(P_\pi^T A_{\mathcal{G}}^l P_\pi) = P_\pi C A_{\mathcal{G}}^{l+1}(P_\pi C)^T \tag{25}$$

$$= P_\pi C A_{\mathcal{G}}^{l+1} C^T P_\pi^T \tag{26}$$

$$= P_\pi h_A(A_{\mathcal{G}}^l) P_\pi^T \tag{27}$$

Equation 25 follows from Proposition 2.

## A.5 PROPOSITION 5

Let denote $L_{\mathcal{G}} = D_{\mathcal{G}} - A_{\mathcal{G}}$ and $L_{\mathcal{H}} = D_{\mathcal{H}} - A_{\mathcal{H}}$ the unnormalized graph Laplacians of the original graph and the spanning supergraph, respectively.

Let define the matrix $L_{\mathcal{S}} := L_{\mathcal{H}} - L_{\mathcal{G}}$. We can interpret $L_{\mathcal{S}}$ as a perturbation of $L_{\mathcal{G}}$.

By Weyl's inequality, we have $|\lambda_{\mathcal{G}}(i) - \lambda_{\mathcal{H}}(i) \leq ||L_{\mathcal{S}}||_2$.

From here, we have:

$$||L_{\mathcal{S}}||_2 = \sqrt{\rho(L_{\mathcal{S}}^T L_{\mathcal{S}})} \tag{28}$$

$$\leq \sqrt{||L_{\mathcal{S}}^T L_{\mathcal{S}}||_\infty} \tag{29}$$

$$\leq \sqrt{||L_{\mathcal{S}}^T||_\infty ||L_{\mathcal{S}}||_\infty} \tag{30}$$

$$= ||L_{\mathcal{S}}||_\infty \tag{31}$$

$$= max_{1 \leq i \leq n}(\sum_j L_{\mathcal{S}i,j}) \tag{32}$$

$$= max_{1 \leq i \leq n}(2\sum_j S_{i,j}) \tag{33}$$

Equation 28, where $\rho()$ denotes the spectral radius, follows the definition of the spectral norm on real-valued matrices. Equation 29 follows from the fact that the spectral radius is upperbouded by any consistent matrix norm. By the sub-multiplicativity of the matrix norm and the symmetry of $L_{\mathcal{S}}$, we get Equation 30 □

### A.6 Proposition B

Let's define the normalized coarsening matrix as $\boldsymbol{R} := \boldsymbol{N}^{-1/2}\boldsymbol{C}$.

We remark that $\boldsymbol{R}$ is a semi-orthogonal matrix, hence $\boldsymbol{R}^T\boldsymbol{R} = \boldsymbol{I}_K$, and that $\boldsymbol{R}\boldsymbol{R}^T = \boldsymbol{C}\boldsymbol{C}^\mp = \boldsymbol{H}$, where $\boldsymbol{H}$ is the projection matrix $\boldsymbol{A}_{\mathcal{H}}^l = \boldsymbol{H}\boldsymbol{A}_{\mathcal{G}}^l\boldsymbol{H}$.

Let $\boldsymbol{L_W}$ and $\boldsymbol{L_{\mathcal{H}}}$ be the normalized Laplacians of the weighted coarsen and lifted graphs, respectively.

We remark that $\boldsymbol{L_W} = \boldsymbol{R}^T\boldsymbol{L_{\mathcal{H}}}\boldsymbol{R}$, and $\boldsymbol{L_{\mathcal{H}}} = \boldsymbol{R}\boldsymbol{L_W}\boldsymbol{R}^T = \boldsymbol{R}\boldsymbol{R}^T\boldsymbol{L_{\mathcal{H}}}\boldsymbol{R}\boldsymbol{R}^T$.

From now on, we reproduce here a proof given by Jin et al. (2020b).

Consider the eigenvalue equation:

$$\boldsymbol{L_W}\boldsymbol{u} = \lambda_i\boldsymbol{u}_i \tag{34}$$

$$\boldsymbol{R}^T\boldsymbol{L_{\mathcal{H}}}\boldsymbol{R} = \lambda_i\boldsymbol{u}_i \tag{35}$$

$$\boldsymbol{R}\boldsymbol{R}^T\boldsymbol{L_{\mathcal{H}}}\boldsymbol{R} = \lambda_i\boldsymbol{R}\boldsymbol{u}_i \tag{36}$$

$$\boldsymbol{R}\boldsymbol{R}^T\boldsymbol{L_{\mathcal{H}}}\boldsymbol{R}\boldsymbol{R}^T\boldsymbol{R} = \lambda_i\boldsymbol{R}\boldsymbol{u}_i \tag{37}$$

$$\boldsymbol{L_{\mathcal{H}}}\boldsymbol{R} = \lambda_i\boldsymbol{R}\boldsymbol{u}_i \tag{38}$$

Thus, $\boldsymbol{L_{\mathcal{H}}}$ contains all the eigenvalues of $\boldsymbol{L_W}$, with $\boldsymbol{R}$ acting as an eigenvector lifting operator.

We observe that $\boldsymbol{I}_{n^l} - \boldsymbol{L_{\mathcal{H}}}$ is at most a rank $n$ matrix because equivalent nodes have the same edge weights in $\boldsymbol{I}_n - \boldsymbol{L_{\mathcal{H}}}$. So, $\boldsymbol{I}_n - \boldsymbol{L_{\mathcal{H}}}$ has eigenvalue 0 with multiplicity $n - K$, and correspondingly, $\boldsymbol{L_{\mathcal{H}}}$ has as much multiplicity of eigenvalue 1.

## B Models

### B.1 Preprocessing

#### B.1.1 Intuitive interpretation

We present a hierarchical graph generative method that progressively generates finer graphs from coarser ones through a series of conditional generation steps. The conditional generation steps are trained and operate on the outputs of a sequence of coarsening and lifting operations, which, in a preprocessing step, transform an input graph to coarser versions of it.

We will now describe the sequence of coarsening and lifting operations. Coarseness level $l = 0$ represents the finer level, $\mathcal{G}^0$ which is the original input graph, and level $l = L$ the coarsest graph level. At level $l$, we coarsen the graph $\mathcal{G}^l$ to produce the graph $\mathcal{G}^{l+1}$ at level $l + 1$. We then lift the coarse graph $\mathcal{G}^{l+1}$ producing the graph $\mathcal{H}^l$, which is a spanning supergraph of $\mathcal{G}^l$, i.e., $\mathcal{H}^l = \mathcal{G}^l + \mathcal{S}$, where $\mathcal{S}$ is a set of additional edges. Thus, at each level $l$, we obtain a pair of graphs $(\mathcal{G}^l, \mathcal{H}^l)$. At the coarsest level $L$, we use the complete graph $\mathcal{K}$, i.e., the graph with edges on all node pairs, as the spanning supergraph. We present a pseudo-code algorithm in Algorithm 1.

We create a coarse graph by learning node partitions and merging nodes from the same partition into a single parent node. Thus, the number of partitions $K$ in a graph at level $l$ corresponds to the number of nodes $n^{l+1}$ in the coarsened graph at level $l + 1$. Coarse nodes are connected if at least one pair of their corresponding child nodes is connected. Each coarse node is annotated with the number of child nodes it contains. In the reverse process, we lift a coarse graph to a finer graph by expanding each coarse node into a clique, with the number of nodes matching the annotation of the coarse node. If two coarse nodes are connected, a biclique is established between their constituent nodes. Note that lifting a graph does not invert the coarsening operation, i.e., it does not recover the graph before coarsening; instead, the process yields a spanning supergraph of it. Figure 2 illustrates a coarsening-lifting procedure step.

To learn the partitions, we propose a new partitioning method, called Gamma-Min, that minimizes the Gamma Index of the spanning supergraph; the Gamma Index indicates the graph sparsity. In

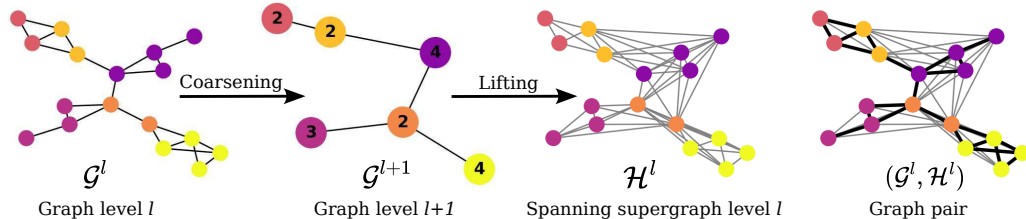

Figure 2: Coarsening and lifting. Graph level $l$: The color represents the node partitions. By coarsening, we obtain the graph at level $l + 1$. Graph at level $l + 1$: Nodes in a given partition at level $l$ have been pooled together. The node attribute indicates the number of child nodes in each coarse node. Lifting: Child nodes from the same parent node or from connected parent nodes are connected in the lifted graph. The lifted graph $\mathcal{H}^l$, which is unannotated, is a spanning supergraph of $\mathcal{G}^l$. Graph pair: The process produces a graph pair for each coarseness level.

section 3, we show that the spanning supergraphs produced by our method preserve the spectral properties of the coarse graph, and we provide an upper bound on the spectral discrepancy between the graph $\mathcal{G}^l$ and its spanning supergraph $\mathcal{H}^l$. We use the spanning supergraph both as a conditioning structure in our hierarchy of generative models, enforcing the preservation of the global spectral information captured at the coarser level, and as a sparse structure for efficient message-passing. By minimizing the Gamma Index of the spanning supergraph, we increase the sparsity of the supergraph structure. This is an important feature of our method since it results in a corresponding reduction of the computational complexity of the generative models that we apply over supergraphs.

---

**Algorithm 1:** Hierarchical Preprocessing

**Data:** $\mathcal{D}_{in} = \mathcal{D}^0 = \{\mathcal{G}_i^0\}_{i=1}^N$, where $\mathcal{G}_i^0 = (\boldsymbol{A}_i^0, \boldsymbol{X}_i^0)$
1 Set $\mathcal{D}_{out} = \{\}$
2 **for** $l = 0$ *to* $L - 1$ **do**
3     Train partitioning model $C_\theta^l$ on $D^l$
4     Set $\mathcal{D}^{l+1} = \{\}$
5     **for** $\mathcal{G}_i^l$ *in* $\mathcal{D}^l$ **do**
6         $\boldsymbol{C}_i = C_\theta^l(\boldsymbol{A}_i^l)$    *Partitioning*
7         $\mathcal{G}_i^{l+1} = (\boldsymbol{X}_i^{l+1}, \boldsymbol{A}_i^{l+1}) = \text{Coarsen}(\mathcal{G}_i^l, \boldsymbol{C}_i))$
8         $\mathcal{H}_i^l = \text{Lift}(\mathcal{G}_i^{l+1})$
9         Pair $(\mathcal{G}_i^l, \mathcal{H}_i^l)$ in $\mathcal{D}^l$
10         $\mathcal{D}^{l+1}$.append($\mathcal{G}_i^{l+1}$)
11     **end**
12     $D_{out}$.append($\mathcal{D}^l$)
13 **end**
14 **for** $\mathcal{G}_i^L$ *in* $\mathcal{D}^L$ **do**
15     Pair $(\mathcal{G}_i^L, \mathcal{K})$
16 **end**
17 $D_{out}$.append($\mathcal{D}^L$)
**Result:** $D_{out} = \{(\mathcal{G}_i^l, \mathcal{H}_i^l)_{i=1}^N\}_0^L$

---

### B.2 RECONSTRUCTION OF THE ASSIGNMENT MATRIX AT GENERATION

During generation, it is necessary to lift the generated graph $\hat{\mathcal{G}}^l$. However, the corresponding assignment matrix is not available directly, but we can recover it (up to a permutation) uniquely from its annotation vector $\boldsymbol{n}_{\hat{\mathcal{G}}}^l$.

To achieve this, we define the node index vector $\boldsymbol{i}_{\hat{\mathcal{G}}^{l+1}} = [1, ..., n]$. Using this, we construct $\boldsymbol{c}$ by repeating each index $i_j$ a number of times equal to the corresponding entry $n_j$. This operation corre-

sponds to the PyTorch function REPEAT_INTERLEAVE($i$, $n$). By one-hot encoding $c$, we eventually recover the assignment matrix $\hat{C}_{\hat{\mathcal{G}}^{l+1}}$.

With $\hat{C}_{\mathcal{G}^{l+1}}$, we can then lift $\hat{\mathcal{G}}^{l+1}$ to $\hat{\mathcal{H}}^l$ using the formula: $A_{\hat{\mathcal{H}}}^l = \hat{C}_{\hat{\mathcal{G}}^{l+1}} \tilde{A}_{\hat{\mathcal{G}}}^{l+1} \hat{C}_{\hat{\mathcal{G}}^{l+1}}^T - I_n$.

### B.3 GAMMA-MIN

#### B.3.1 MODEL ARCHITECTURE

We train the Gamma-Min partitioning using a GNN, which includes three layers of MPNN identical to the one used for discrete diffusion and detailed in Appendix B.4.2, followed by three Transformer blocks (Vaswani et al., 2017). We linearly project the network outputs to the number of partitions before applying hardmax.

#### B.3.2 GAMMA-INDEX

We present in Table B.3.2 all the Gamma-Index used for experimentation and their training time.

| Dataset | level | $n_{\max}$ | Reduc. | $\gamma_{\text{data}}$ | $\gamma_{\mathcal{H}}$ | Time (s) |
|---------|-------|------------|--------|------------------------|------------------------|----------|
| Zinc250k | 0 | 38 | 4 | 0.094 | 0.247 | 991 |
| SBM20k | 0 | 194 | 3 | 0.083 | 0.213 | 661 |
| SBM20k | 1 | 67 | 3 | 0.181 | 0.410 | 478 |
| SBM20k | 2 | 23 | - | - | - | - |
| GitStar | 0 | 957 | 5 | 0.016 | 0.053 | 4794 |
| GitStar | 1 | 192 | 4 | 0.059 | 0.195 | 1573 |
| GitStar | 2 | 48 | 3 | 0.160 | 0.378 | 1491 |
| GitStar | 3 | 16 | - | - | - | - |
| Reddit12k | 0 | 1499 | 3 | 0.005 | 0.017 | 5891 |
| Reddit12k | 1 | 500 | 3 | 0.014 | 0.059 | 1516 |
| Reddit12k | 2 | 167 | 3 | 0.048 | 0.197 | 1281 |
| Reddit12k | 3 | 56 - | | - | - | - |

Table 4: Gamma-Index for all datasets and all levels

#### B.3.3 TRAINING AND HYPERPARAMETERS

Table 5: Hyperparameters identical for all experiments

| | |
|---|---|
| Layers in MLPs | 3 |
| MPNN layers | 3 |
| Transformer Blocks | 3 |
| Learning rate | 0.0002 |
| Optimizer | Adam |
| Betas parameters for Adam | (0.9, 0.999) |
| Extra feature: eigen features | True |
| Extra feature: number of nodes | True |
| Layers in MLPs | 3 |

#### B.3.4 EVALUATION

In our experiments, we compare our coarsening model with two baselines: DiffPool (Ying et al., 2018) and MinCut (Bianchi et al., 2020). We evaluate graph sparsity in the spanning supergraph at the finest level, using the described coarsening approach and parameters. All three models rely on an assignment matrix $C$ parameterized by a Graph Neural Network (GNN), with the same GNN architecture used across models. The key difference lies in their training objectives: DiffPool minimizes a link prediction objective, MinCut minimizes a continuous relaxation of the normalized minCUT problem, our GammaMin minimizes the Gamma-Index of the spanning supergraph. Both baseline

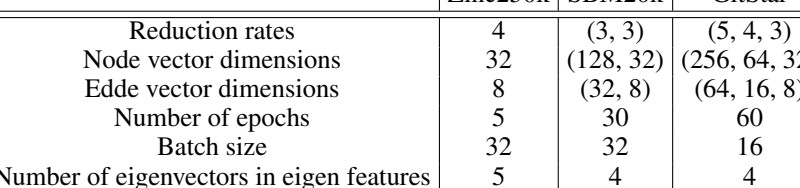

Table 6: Hyperparameters depending on the dataset

|  | Zinc250k | SBM20k | GitStar | Reddit |
|---|---|---|---|---|
| Reduction rates | 4 | (3, 3) | (5, 4, 3) | (3, 3, 3) |
| Node vector dimensions | 32 | (128, 32) | (256, 64, 32) | (256, 128, 32) |
| Edde vector dimensions | 8 | (32, 8) | (64, 16, 8) | (64, 32, 8) |
| Number of epochs | 5 | 30 | 60 | 40 |
| Batch size | 32 | 32 | 16 | 8 |
| Number of eigenvectors in eigen features | 5 | 4 | 4 | 20 |

models require additional regularization terms to avoid degenerate solutions, such as assigning most nodes to a single cluster. Our model do not need such regularization.

### B.4 HIERARCHICAL EQUIVARIANT DISCRETE DIFFUSION

#### B.4.1 FORWARD NOISING PROCESS

Austin et al. (2021) proposed many options for the noising model leading to a fixed limit distribution $q(x^T|x^0) = q(x^T)$. We use the standard categorical transition matrix $\boldsymbol{Q}^t = \alpha^t I + (1 - \alpha^t)\boldsymbol{1}m'$, where $m'$ is the row vector of the limit distribution.

We use the marginal distribution of the node attributes as the limit distribution for the node attributes. For the edges, we use the 'non-edge' category as an absorbing state (see Austin et al. (2021)) so that all the mass of the limit distribution is on the 'non-edge' attribute.

For $\alpha$, we employ the standard cosine schedule $\alpha = \cos(0.5\pi(t/T + s)/(1 + s))^2$, with a small $S$.

#### B.4.2 DENOISING NETWORKS

We use standard message-passing neural networks (MPNN) for our denoising models, with $L$ layers following these two equations:

$$\boldsymbol{e}_{i,j}^{l+1} = \text{LayerNorm}(f_{\text{edge}}([\boldsymbol{x}_i^l, \boldsymbol{x}_j^l, \boldsymbol{e}_{i,j}^l]) \tag{39}$$

$$\boldsymbol{x}_i^{l+1} = \text{LayerNorm}\left(\boldsymbol{x}_i^l + \sum_{j \in \mathcal{N}(i)} f_{\text{node}}([\boldsymbol{x}_i^l, \boldsymbol{x}_j^l, \boldsymbol{e}_{i,j}^l])\right), \tag{40}$$

where $[\cdot, \cdot]$ is the concatenation operator and the $f$ functions are 3-layers MLPs. The neighborhood $\mathcal{N}(i)$ on the right-hand side of equation 40 defines the adjacent node in the spanning supergraph.

The last layer outputs are projected to corresponding dimensions. We enforces edge symmetry with $\boldsymbol{e}_{i,j} = (\boldsymbol{e}_{i,j} + \boldsymbol{e}_{j,i})/2$. We ensure that the output can be interpreted as probabilities by applying either the softmax of the sigmoid function.

We use the same number of vector dimensions for the node and edge representations and for the hidden sizes of the MLPs. The dimensions used for each experiment are reported hereunder.

### B.5 SAMPLING

diffusion $p(\mathcal{G}_l^{t-1}|\mathcal{G}_l^t, \mathcal{H}_l)$, which We model a single denoising step as a product over nodes and edges:

$$p(\mathcal{G}_l^{t-1}|\mathcal{G}_l^t, \mathcal{H}_l) = \prod_{i=1}^{n} p_\theta(\boldsymbol{x}_i^{l,t-1}|\mathcal{G}_l^t, \mathcal{H}_l) \prod_{j=1}^{m_{\mathcal{H}}} p_\theta(\boldsymbol{e}_i^{l,t-1}|\mathcal{G}_l^t, \mathcal{H}_l) \tag{41}$$

We approximate this probability distribution by marginalizing over the network prediction as in standard discrete diffusion. However, in generation, we do not have access to $\mathcal{H}^l$. Instead, we use

the graph $\hat{\mathcal{H}}_l$ obtained by lifting the graph $\hat{\mathcal{G}}_{l+1}$ generated at the coarser level. The marginalization becomes:

$$p_\theta(\boldsymbol{e}_i^{t-1}|\mathcal{G}_l^t, \mathcal{H}_l) = \sum_{d=1}^{d_e} p(\boldsymbol{e}_i^{t-1}|\boldsymbol{e}_i = d, \mathcal{G}_l^t, \hat{\mathcal{H}}_l)p_\theta(\boldsymbol{e}_i = d|\hat{\mathcal{H}}_l). \tag{42}$$

### B.5.1 TRAINING AND HYPERPARAMETERS

Table 7: Hyperparameters fixed for all experiments

| | |
|---|---|
| MPNN layers | 4 |
| Layers in MLPs | 3 |
| Diffusion steps | 500 |
| Learning rate | 0.0002 |
| Optimizer | Adam |
| Betas parameters for Adam | (0.9, 0.999) |
| Extra feature: eigen features | True |
| Extra feature: number of nodes | True |
| Layers in MLPs | 3 |

Table 8: Hyperparameters depending on the dataset

| | Zinc | SBM20k | GitStar | Reddit |
|---|---|---|---|---|
| Reduc. rates | 4 | (3, 3) | (5, 4, 3) | (3, 3, 3) |
| Node vect. dim. | (256, 64) | (64, 64, 64) | (64, 64, 64, 64) | (64, 64, 64, 64) |
| Edge vect. dim. | (64, 16) | (32, 64, 64, 64) | (64, 64, 64) | (64, 64, 64, 64) |
| # epochs | (40, 40) | (125, 250, 250) | (150, 150 ,150, 150) | (100, 100 ,100, 100) |
| Batch size | (64, 64) | (16, 64, 64, 64) | (16, 64, 64, 64) | (16, 64, 64, 64) |
| # of eigenvect. | 5 | 4 | 4 | 20 |

### B.5.2 EXTRA FEATURES

We use three extra features: eigen features, graph size, and partition size. All these features are concatenated to the input node attributes.

**Eigen features**   First, we use the eigenvectors associated with the $k$ lowest eigenvalues of the graph normalized Laplacian of the spanning supergraph, which are (up to normalization) identical to the eigenvectors of the corresponding coarse graph, except in level $L$, where we recompute the eigenvector of the noisy graph at each training step. Otherwise, and unlike other denoising models, the computation of the eigenvectors is a single preprocessing step.

**Graph size**   We use the graph size represented as a ratio between the size of the current graph and the largest graph in the dataset $n/n_{max}$. We concatenate the (same) value to all nodes in a graph.

**Partition size**   Similarly, we use the partition size, which is represented as a ratio between its size and the largest partition in the dataset. All nodes in the same partition (in a partition with the same number of nodes) have the same value.

## C   RELATED WORK

### C.1   GRAPH COARSENING

We further explain why node-dropping methods and contraction methods are not suitable choices for hierarchical generative models.

Node-dropping methods (Gao & Ji, 2019; Lee et al., 2019) coarsen a graph by removing specific nodes and their associated edges , either one-by-one or by blocks. However, node-dropping induces

a node ordering, which breaks equivariance. Furthermore, reversing the process typically leads to node-wise or block-wise autoregressive models, similar to existing approaches, such as Liao et al. (2019).

Contraction methods (Loukas, 2019; Diehl, 2019) coarsen graphs by merging nodes through either edge contraction, where the endpoints of an edge are merged, or neighborhood contraction, where all adjacent nodes of a given node are merged. However, these methods require a specific contraction ordering, which breaks equivariance, to ensure that no node belongs to multiple contracted clusters. Additionally, contraction methods offer limited control over the coarsening factor, i.e., the ratio of nodes before and after coarsening.

## D  RELATED WORKS

### D.1  GENERATIVE MODELS

One of the main challenges in graph representation and generation is that a graph can be represented in multiple ways. There can be up to $n!$ different representations of the same graph, resulting from the $n!$ possible node permutations. Two main approaches address the multiplicity of equivalent representations: models that operate sequentially and equivariant models.

**Sequential models**  generate graphs by auto-regressively adding nodes, edges, or subgraphs. To limit the number of different sequences that represent a single graph, most of these models use a Breadth-First Search (BFS) approach (You et al., 2018; Shi et al., 2020; Luo et al., 2021; Liao et al., 2019; Kong et al., 2023). While canonical representations exist for specific domains, e.g. canonical SMILES for molecular graphs (Gómez-Bombarelli et al., 2018; Kusner et al., 2017), methods based on general graph canonization (Goyal et al., 2020) fail for large graphs (see experiments in Bergmeister et al. (2024)). Subgraph aggregation (Jin et al., 2018; 2020a), sometimes described as hierarchical, falls into this category. It requires listing the set of all possible substructures and connections between them, which is feasible only for some specific applications such as molecular graphs.

**Equivariant models**  address the node permutation issue by ensuring a unique computational graph for all possible instantiations of the same object. These models have been developed within various generative frameworks such as GANs (Krawczuk et al., 2021; Martinkus et al., 2022) or Normalizing Flows (Madhawa et al., 2019; Zang & Wang, 2020; Liu et al., 2019). Recently, equivariant denoising models used in score-based diffusion (Yang et al., 2019; Jo et al., 2022), discrete diffusion (Haefeli et al., 2022; Vignac et al., 2023), diffusion bridges (Jo et al., 2024), and Flow Matching (Eijkelboom et al., 2024) have significantly improved graph generation for small graphs. While less known, equivariant quantized auto-encoders have also demonstrated competitive performance (Boget et al., 2024; Nguyen et al., 2024). However, equivariant models are not without limitations. They operate by producing predictions for all node pairs and rely on dense graph representations, which prevents them from scaling to large graphs.

**Sparse Equivariant Model**  SparseDiff (Qin et al., 2024) and EDGE Chen et al. (2023) are recent diffusion models that address scalability challenges in equivariant models. Both models share similar objectives and generative framework as ours. We comparative analysis to highlight their differences from ours.

Both models construct a sparse structure by selecting a subset of "active" nodes, with 'active edges' defined by their induce complete graph. SparseDiff randomly selects "active" nodes, while EDGE determines them based on predicted changes in degree. As a result, in both models, for the same graph, the sparse structure varies at each iteration. In contrast, our model maintains a fixed structure of 'active edges' established by the spanning supergraph.

The sparse strategies employed by SparseDiff and EDGE lead to a number of function evaluations per node and edge, which is critical for denoising models and represents a fraction of the total diffusion steps. SparseDiff compensates for this by increasing function evaluations, denoising the graph in blocks of nodes and edges, ultimately considering all node pairs. This approach results in a number of function evaluations (NFE) that is inversely proportional to the edge fraction in each

block. EDGE does not compensate for excluded edges, which, we assume, is the reason for its comparatively low generative performance (see 5).

At generation, we still leverage our fixed graph structure. In our model, the additional computational cost arises from generating graphs at lower levels. However, since these graphs are significantly smaller and computational cost scales quadratically with the number of nodes, this cost is comparatively small.

**Hierarchical models** A couple of works follow a similar hierarchical approach to ours, but differ in their coarsening and generative strategies. HiGen (Karami, 2024) coarsens the graph using a modularity objective to partition the graph into communities. The community-based partitions often produce dense coarse graphs, which strongly limits the effectiveness of the strategy, both in terms of the information extracted from the coarsening and the resulting sparsity used for generation. Moreover, the model leverages an autoregressive method to generate graphs at each level, which is inefficient for large graphs.

Bergmeister et al. (2024) use a local coarsening scheme involving edge or neighborhood contraction. To prevent the coarsening from pooling a child node into two parent nodes, by contracting two of its adjacent edges, they sample a different contraction sequence at each training iteration. Moreover, the method requires a low reduction rate - set to a maximum of 0.3 - necessitating sequences with many coarseness levels. In our experiments, we were not able to generate large graphs in reasonable time with this method (see Section 5). In contrast to these hierarchical models, our method uses a single coarsening procedure in a preprocessing step, a small number of levels, and, last but not least, maintains equivariance.

# E EVALUATION

## E.1 DOWNLOADS

To download the Stochastic Block Model 20k (SBM20k) in the pytorch geometric Dataset format:

https://drive.switch.ch/index.php/s/t5I9N8rDQCfMIVX

To download the splits between training and test sets used in our experiments:

- SBM20k: https://drive.switch.ch/index.php/s/zhlXUa4mUKyCP3G

- GitStar: https://drive.switch.ch/index.php/s/ADn014uV44Kwcbj

- Reddit12k https://drive.switch.ch/index.php/s/xIS3DMY2eUCzN8c

## E.2 ABLATION STUDY

We present an ablation study to analyze the effect of the number of hierarchical levels $L$. Models were trained with 1 to 5 levels on SBM20k and 2 to 7 levels on GitStar, keeping all other hyperparameters fixed.

We fixed the total reduction in the number of nodes (relative to the $n_{\max}$), the number of node in the largest graph. We set this reduction $R$ to 16 for SBM20k and 64 for Gitstar. Since the total reduction is fixed, the more steps the smaller the reduction at each step. We used a fixed reduction rate corresponding to that $R = n_{\max}^{(\frac{1}{L})}$. Hence, the maximum number of nodes at each level is given by: $n_{\max}^{l+1} = n_{\max}^l / R$.

The results are reported in Figures E.2 and E.2.

We make four key observations:

1. The Gamma Index of the spanning supergraph decreases as the number of levels increases at the data level. Consequently, we were unable to train models with only one or two levels on GitStar.

2. All hierarchical models outperform their non-hierarchical counterpart on SBM20k.

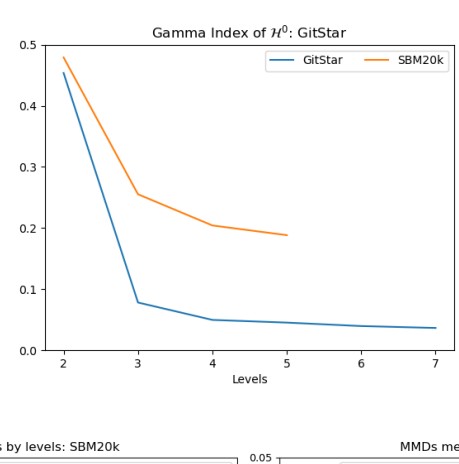

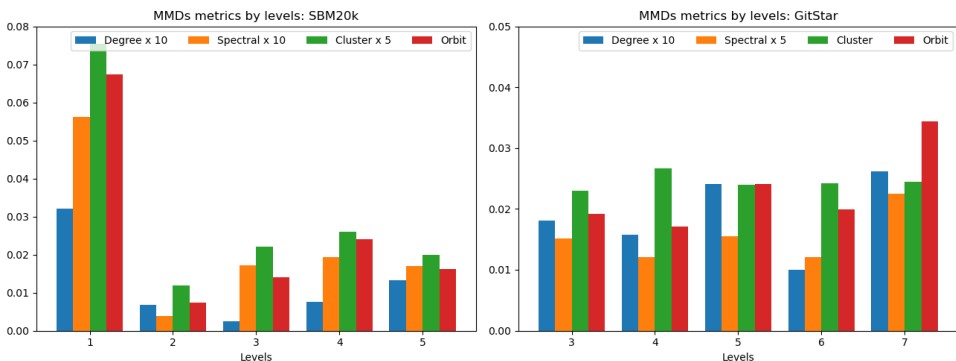

Figure 3: Effect of the number of levels on the generative performance.

3. The Degree MMD metric exhibits relatively large variations that appear unrelated to the number of levels. We are currently unable to explain these variations.

4. On SBM20k, except the non-hierarchical model, performance slightly decreases with the number of levels across all other metrics.

5. No similar relationship between performance and the number of levels is observed on Git-Star.

In conclusion, hierarchical models consistently outperform their non-hierarchical counterpart. However, we do not find a clear or systematic relationship between the number of levels and overall model performance.

### E.3 CONDITIONAL GENERATION

In conditional generation, we generate graphs $\hat{\mathcal{G}}$ conditionally to the spanning supergraph $\mathcal{H}$. In our experiment, we sample a graph from the validation set to serve as the reference graph $\mathcal{G}_{ref}$. The conditional task involves generating graphs structurally similar to $\mathcal{G}_{ref}$ by conditioning on its spanning supergraph $\mathcal{H}_{ref}$. Since we directly sample from an actual spanning supergraph, only the finer-level model $p_\theta(\mathcal{G}^0|\mathcal{H}^0)$ is needed for conditional generation.

We use the spectral distance to measure the distance between the reference graph and the conditionally generated graphs.

To evaluate similarity, we use the spectral distance between the reference graph and the conditionally generated graphs. The spectral distance between graphs $\mathcal{G}_1$ and $\mathcal{G}_2$ is defined as $\sum_{i=1}^{K} |\lambda_{\mathcal{G}_1}(i) - \lambda_{\mathcal{G}_2}(i)|$, where $\lambda(i)$ is the $i^{\text{th}}$ eigenvalue of the Laplacian sorted in non-decreasing order. To ensure that we can compute this distance between graphs of different sizes, we compute this distance only on the $K$ smallest eigenvalues.

Table 9: Spectral distances

| Dataset | Test set | Cond. gen. | Ratio |
|---|---|---|---|
| SBM20k | $4.57 \pm 1.55$ | $0.49 \pm 0.06$ | $0.114 \pm 0.024$ |
| Reddit | $3.57 \pm 1.10$ | $0.57 \pm 0.31$ | $0.161 \pm 0.069$ |

We computed the spectral distance between the reference graph and 100 conditionally generated graphs, and compared it with the distances between the reference graph and graphs from the test set. This experiment was repeated 10 times with different reference graphs. Table 9 shows the average distances and ratios. On average, the generated graphs were approximately 10 times closer to the reference graph than those in the test set, demonstrating the effectiveness of our model's conditional setting. We provide illustration of conditionally generated graph in Appendix F.

### E.4 DATASETS

**Zinc250** The Zinc250k dataset is a subset of the Zinc database (Irwin et al., 2012). It includes 250,000 molecules with up to 38 heavy atoms of nine types. We used the kekulized representation of this dataset.

**SBM20k** The original Stochastic Block Model is a synthetic dataset made of community graphs, with 2 to 5 communities, each containing 20 to 40 vertices. The intra-community and inter-community edge probability are 0.3 and 0.005, respectively. We create a dataset, SBM20k, with exactly the same characteristics but 20000 instances instead of 200.

**GitStar** Github Stargazers (Rozemberczki et al., 2020) is a collection of 12725 graphs with up to 957 nodes representing social networks. It is part of the TUDatasets, which are benchmark datasets collected from the TU Dortmund University.

**Reddit12k** Reddit(Yanardag & Vishwanathan, 2015) contains graphs extracted from the Reddit networks. It is also part of the TUDataset. We extracted the graphs containing up to 1500 nodes. This results in a dataset that collects 11551 graphs with up to 1499 nodes.

### E.5 EVALUATION PROCEDURE

#### E.5.1 ZINC250K

The benchmark results for Zinc250 are taken from the original paper, except for DiGress and SparseDiff, which we ran ourselfs. We used the Official SparseDiff repository to implement the Zinc250k dataset.

**Spits** We used the test sets provided by (Jo et al., 2022). The metrics are calculated over 10,000 samples from the test sets.

**metrics** We use the Fréchet ChemNet Distance (FCD) (Preuer et al., 2018) and the Neighborhood subgraph pairwise distance kernel (NSPDK) MMD (Costa & Grave, 2010) metrics. FCD assesses the generated molecules in chemical space, while NSPDK MMD evaluates the distribution of the graph structures. In addition to FCD and NSPDK metrics, we include the validity rate without correction as a supplementary evaluation metric. This metric calculates the fraction of valid molecules without valency correction or edge resampling.

**Additional metrics** As additional metrics presented in table E.5.1, we report the uniqueness - the fraction of unique generated molecules - and the novelty - the fraction of unique molecules not in the dataset. All models yield 100% validity with valency correction or resampling.

#### E.5.2 LARGE DATASETS: SMB20K, GITSTAR, REDDIT12K

We split the datasets into a test set with 1000 graphs and a training set with the remaining graphs. We further split the remaining instances between the training and validation sets.

Table 10: Generation results on the **Zinc** dataset.

| Model | Uniqueness↓ | Novelty↓ |
|---|---|---|
| GDSS | $99.64 \pm 0.13$ | $100.00 \pm 0.00$ |
| DGAE | $99.94 \pm 0.03$ | $99.97 \pm 0.01$ |
| DiGress | $99.98 \pm 0.01$ | $99.99 \pm 0.01$ |
| SparseDiff | $100.00 \pm 0.00$ | $100.00 \pm 0.00$ |
| DiscDiff | $99.94 \pm 0.02$ | $99.99 \pm 0.01$ |
| HEDD | $99.99 \pm 0.01$ | $99.99 \pm 0.01$ |

**metrics** We employ the maximum mean discrepancy (MMD) to compare the distributions of graph statistics between generated and test graphs (You et al., 2018). The MMDs are computed over the distributions of degrees (deg.), clustering coefficients (clust.), and the number of occurrences of orbits with up to four nodes (orbit) and the graph spectrum (spect.).

Similar to (Martinkus et al., 2022), we utilize the total variation distance kernel to compute the MMDs. The MMDs are computed by comparing the test set to the same number of generated samples. Due to the slow generation of SparseDiff, the MMDs for Gitstar and Reddit are calculated over 100 graphs of the test set and the same number of generated samples.

**Results on larger samples** Due to the slow generation time from SparseDiff, we used 'only' 100 graphs for the GitStar and Reddit12K datasets in the main text. Here, we provide results for our method (HEDD) with 1000 graphs for future comparison.

Table 11: Generation results on the **Reddit12k GitStar** dataset. Results are rescaled by $10^3$.

| Dataset | degree↓ | clust.↓ | orbit↓ | spect.↓ |
|---|---|---|---|---|
| GitStar | $0.49 \pm 0.21$ | $4.86 \pm 0.99$ | $3.73 \pm 0.94$ | $1.27 \pm 0.32$ |
| Reddit12k | $3.87 \pm 2.35$ | $5.58 \pm 1.37$ | $43.73 \pm 28.81$ | $6.61 \pm 2.38$ |

### E.6 CONFIGURATION OF BENCHMARK MODELS

#### E.6.1 DIGRESS AND SPARSEDIFF

SparseDiff is built upon DiGress to such an extent that it can be viewed as a revised version of DiGress. The official DiGress repository even refers to SparseDiff for training on large graphs. By setting the edge fraction generated at each step to 1, SparseDiff effectively reverts to DiGress. For our experiments, we used this version of DiGress. It explains that both models share identical hyperparameters.

Table 12: Hyperparameters fixed for all experiments

| | |
|---|---|
| learning rate | 0.0002 |
| optimizer | adamw |
| weight decay | 1e-12 |
| diffusion steps | 500 |
| diffusion noise schedule | 'cosine' |
| dropout | 0.1 |
| output y | False |
| scaling layer | False |
| extra features | 'all' |
| eigenfeatures | True |
| edge features | 'all' |
| num. eigenvectors | 8 |
| num. eigenvalues | 5 |
| use charge | False |
| num. degree | 10 |
| positional encoding | False |

Table 13: Hyperparameters depending on the dataset

| SparseDiff | Zinc | SBM20k | GitStar | Reddit |
|---|---|---|---|---|
| Epochs | 20 | 20 | 20 | 20 |
| batch size | 16 | 16 | 8 | 16 |
| n layers | 5 | 4 | 4 | 4 |
| edge fraction | 0.5 | 0.25 | 0.25 | 0.1 |
| de | 64 | 64 | 64 | 64 |
| dx | 256 | 64 | 64 | 64 |
| dy | 64 | 64 | 64 | 64 |
| dim ffe | 128 | 128 | 128 | 64 |
| dim ffx | 256 | 128 | 128 | 64 |
| dim ffy | 256 | 128 | 128 | 64 |

Table 14: Hyperparameters depending on the dataset

| Digress | Zinc | SBM20k |
|---|---|---|
| Epochs | 20 | 20 |
| batch size | 16 | 16 |
| n layers | 5 | 4 |
| de | 64 | 64 |
| dx | 256 | 64 |
| dy | 64 | 64 |
| dim ffe | 128 | 128 |
| dim ffx | 256 | 128 |
| dim ffy | 256 | 128 |

### E.6.2 EDGE

Table 15: Hyperparameters for EDGE

| EDGE | SBM20k | GitStar | Reddit |
|---|---|---|---|
| batch size | 4 | 4 | 4 |
| num iter | 256 | 256 | 256 |
| num workers | 8 | 8 | 8 |
| epochs | 50000 | 50000 | 50000 |
| seed | 0 | 0 | 0 |
| loss type | vb ce xt prescribred st | vb ce xt prescribred st | vb ce xt prescribred st |
| diffusion steps | 512 | 512 | 512 |
| diffusion dim | 64 | 64 | 64 |
| dropout rate | 0.1 | 0.1 | 0.1 |
| num heads | [8, 8, 8, 8, 1] | [8, 8, 8, 8, 1] | [8, 8, 8, 8, 1] |
| arch | TGNN degree guided | TGNN degree guided | TGNN degree guided |
| noise schedule | linear | linear | linear |
| optimizer | adam | adam | adam |
| lr | 0.0001 | 0.0001 | 0.0001 |
| warmup | None | None | None |
| momentum | 0.9 | 0.9 | 0.9 |
| momentum sqr | 0.999 | 0.999 | 0.999 |
| gamma | 0.1 | 0.1 | 0.1 |

We use the training template for large network datasets provided by the official EDGE repository for all the experiments.

# F VISUALIZATIONS

We present visualizations of real and generated graphs for Zinc, GitStar and Reddit12k.

For SBM20k, we present all graphs represented in the preprocessed dataset and their corresponding graph during generation.

We also provide visualizations of conditionally generated graphs, including their reference graph and their conditioning strucuture.

## F.1 ZINC

Figure 4: Comparison of generated molecules with molecules from the Zinc Dataset.

## F.2 GITSTAR

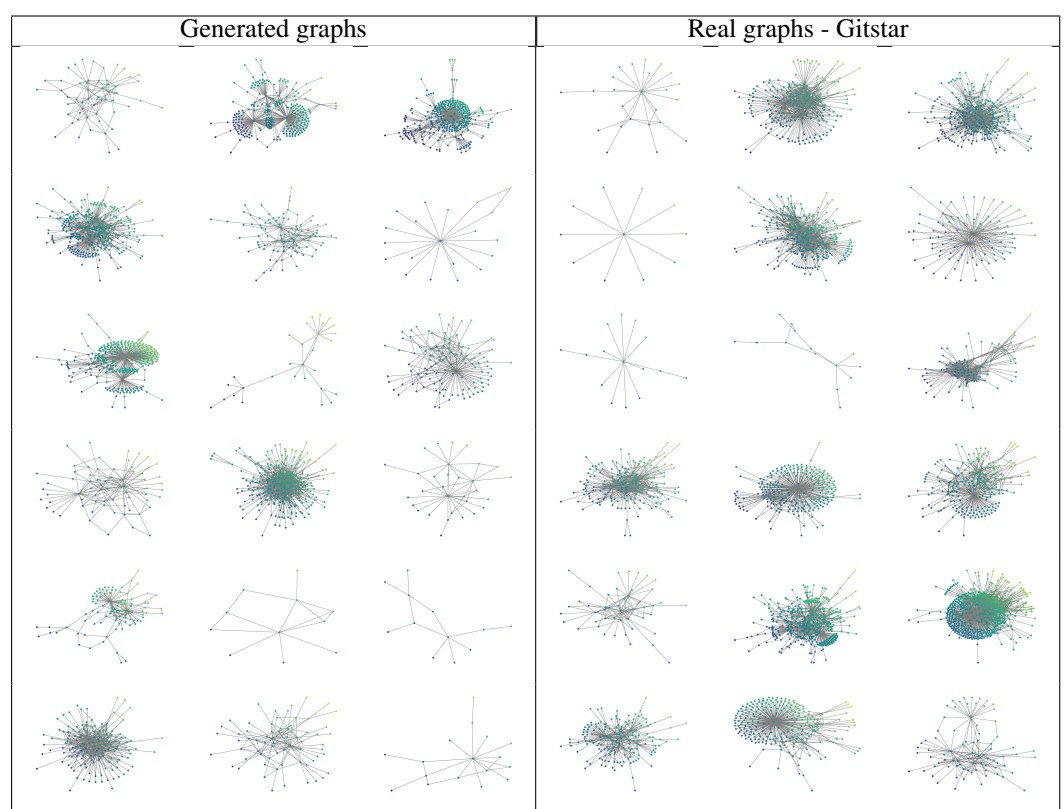

Figure 5: GitStar: Comparison of generated graphs with graphs from the dataset.

## F.3 REDDIT

| Generated graphs | Real graphs - Reddit |
|---|---|

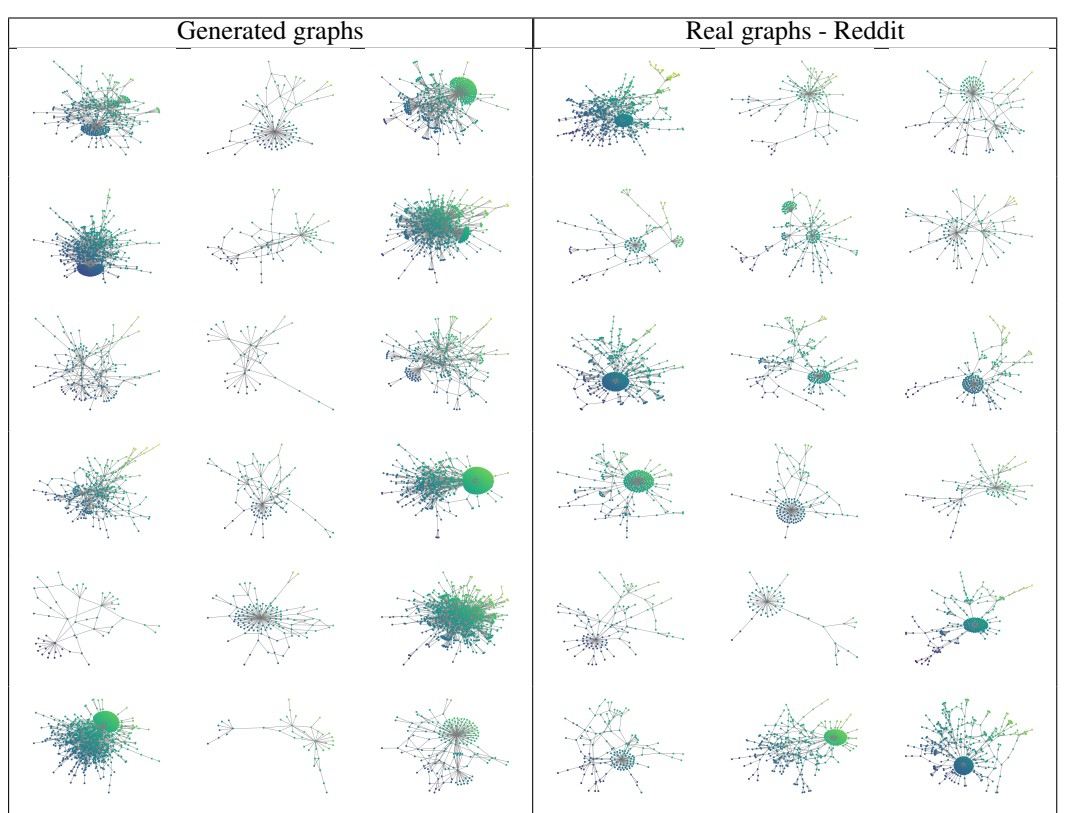

Figure 6: Reddit12k: Comparison of generated graphs with graphs from the dataset.

## F.4 SBM20к

Figure 7: Generated graphs: SBM20k

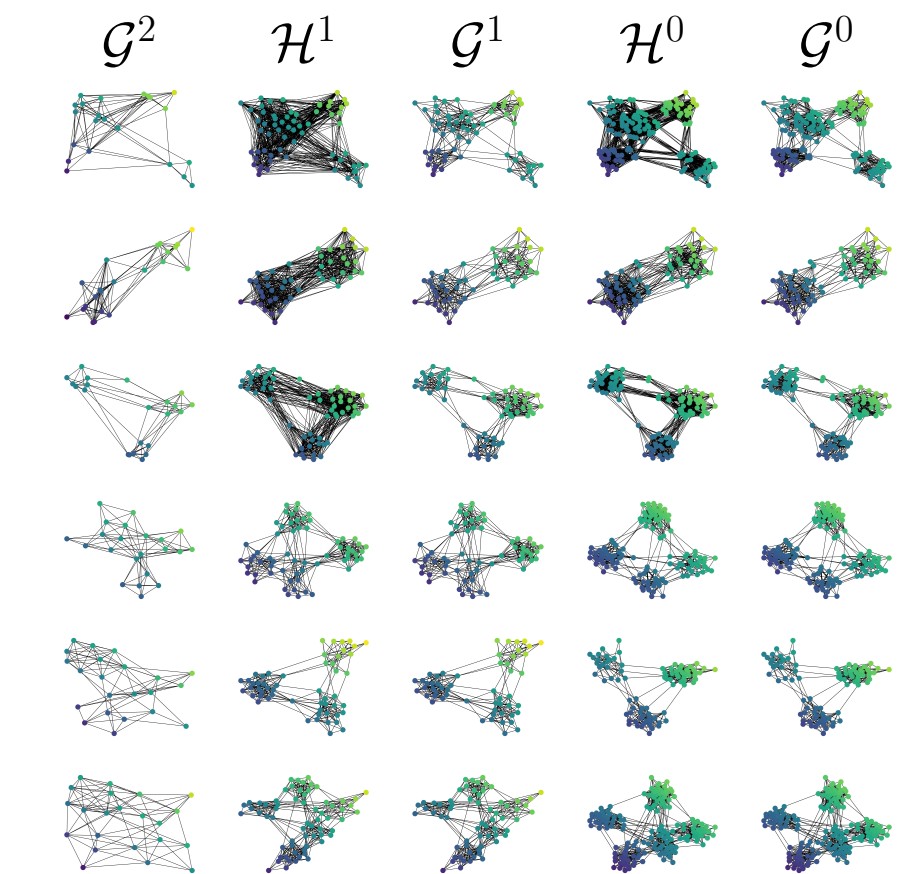

Figure 8: Processed data: SBM20k

## F.5 CONDITIONAL GENERATION

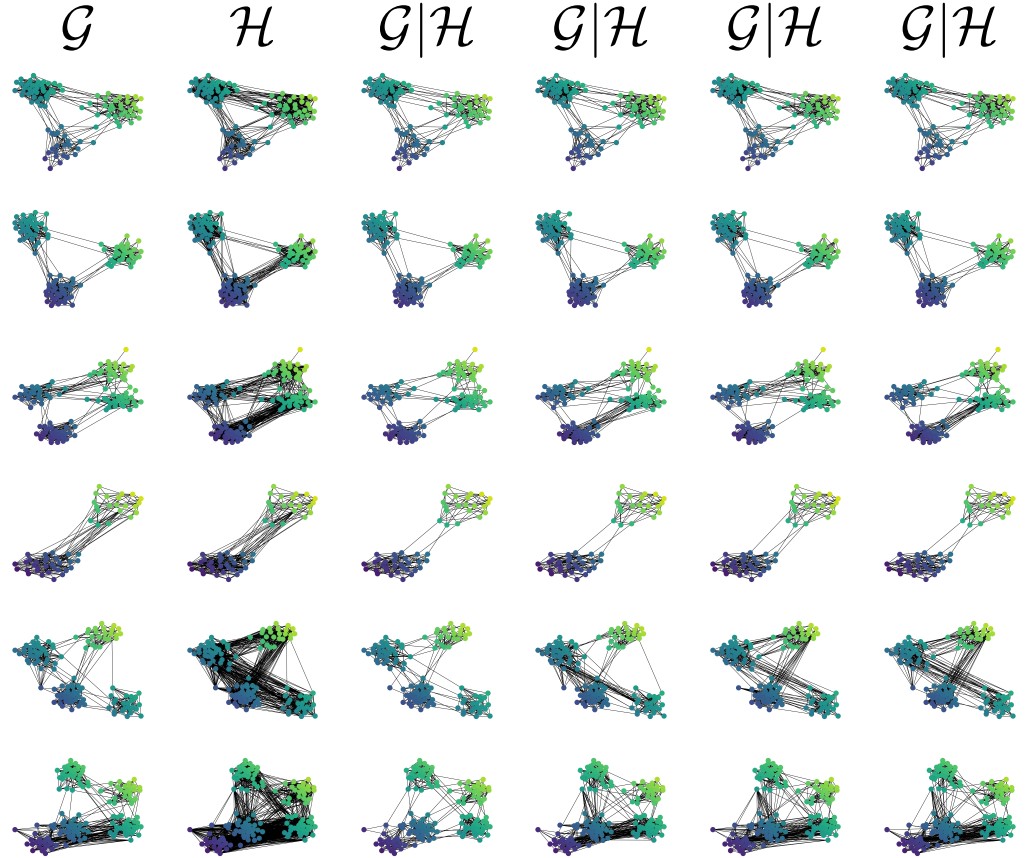

Figure 9: Conditional Generation on SBM20k. First column is the reference graph, the second column its spanning supergraph that serves for the conditional generation, the other columns are conditionally generated graphs (although some are very similar they are all different).

