# OpenReview forum: "HIERARCHICAL EQUIVARIANT GRAPH GENERATION"
_ICLR.cc/2025/Conference — Submitted to ICLR 2025_

### Official Review · Reviewer_F6xy · 2024-10-24

**Soundness:** 3
**Presentation:** 2
**Contribution:** 3
**Rating:** 6
**Confidence:** 4

**Summary:**

The manuscript outlines a new approach to graph generative modeling by addressing challenges such as scalability and maintaining node permutation equivariance. Specifically, current denoising models for graph generation face issues in capturing global graph properties from local interactions. Additionally, while existing equivariant models manage node permutation issues, they often struggle with scalability, particularly when dense graph representations with O(n^2) complexity are involved. To overcome these shortcomings, the authors introduce a novel coarsening-lifting method that generates sparse spanning supergraphs. These supergraphs help in preserving global graph properties and are used as both conditioning structures and sparse layouts for MPNN in generative models. By combining this method with discrete diffusion, the model handles graphs hierarchically, improving the efficiency of generating large graphs.

**Strengths:**

This manuscript proposed a novel hierarchical graph generation method. Firstly, it proposes the gamma-min partitioning method to generate the coarse supergraph G, and then spann the G to a spanning-garph H. With the guidance from G and H, the authors employed the diffusion model for generation. In the experiment, this work was comprehensively evaluated on multiple synthetic and real datasets.

**Weaknesses:**

1. This manuscript proposes a hierarchical equivariant graph generation model. Specifically, the generated object is a 2D graph, and the equivariance is guaranteed on the permutation group by using MPNN as the basic building block. However, equivariance is not a major contribution of this work, so it is somewhat overemphasized in the title of the manuscript. Moreover, it is suggested to illustrate the generated object at first.
2. The manuscript is lengthy, and the experiments are comprehensive. However, some sections do not have a critical correlation with the proposed method, such as the invariance and equivariance (which are not the main contributions and stem from the properties of MPNN itself) and the spectral properties (a long analysis with proof but not directly applied to the generation).
3. In contrast to the equivariant property, the generation model lacks sufficient detail. The authors should describe the generation model more clearly, including how the graph is encoded, how the constraints of $G$ and $H$ are applied to the diffusion layer, and what the loss function is.

**Questions:**

1. In this manuscript, the proposed method focuses only on 2D graphs, and the equivariance correspondingly pertains only to permutations. It is confusing when reading the introduction without a clear definition of the graph category.
2. The graph coarsening process, in some ways, appears similar to node clustering. If I cluster the nodes with their neighbors first (or just combined with some neighbors as the supernode), would that lead to similar results compared to the proposed coarsening method? Maybe some ablation experiments will prove that.
3. In Equation (2), the definition of $W$ should be clarified.
4. It is recommended to adjust the content of the manuscript, delete some unnecessary sections amd introduce more details about the generation model, such as how the guidance of the coarsened and spanned graph is employed, and how the model is supervised.
5. It is suggested to present the related work first before introducing your method.

---

> ### Author Response · Authors · 2024-11-13
>
> Thank you for your thorough review, feedback, and questions. We will first address the points raised under "weaknesses" and then respond to your 'questions'.
>
> 1. Equivariance
>
> We agree that the equivariance of the conditional generative models at each level arises from the use of MPNNs and is not a contribution of our work. However, we believe that preserving equivariance to node permutations within a hierarchical model is a key contribution of our paper. To our knowledge, this is the first model to be simultaneously hierarchical and equivariant.
>
> Regarding the presentation suggestion, we will comeback as soon as possible with a revised version.
>
> 2. Less Relevant Sections
>
> In Section 3, we demonstrate that our coarsening and lifting methods are invariant and equivariant to node permutations. This property does not directly result from the use of MPNNs, as MPNNs are not directly involved in the coarsening or lifting processes.
> Regarding the spectral properties, we will clarify their relevance to our method in the revised paper, though we continue to consider them directly related to our hierarchical approach. Specifically, the spectral distance quantify the information loss between G and H, which is an crucial insight on our hierarchical method.
>
> 2. Model Presentation
>
> We will improve the overall presentation of the paper and provide more detailed explanations of the model, as you suggested.
>
> Questions:
>
> 1. Graph Object
>
> We will ensure that the revised manuscript clearly specifies that we are generating graphs in the form of a set of nodes and a set of edges, and no additional objects.
>
> 2. Comparison with Other Graph Clustering
>
> Graph coarsening in our approach indeed involves graph clustering (which we refer to as partitioning). The alternative approach you suggest (merging adjacent or neighboring nodes) is often called edge or neighborhood contraction. We mention these methods briefly in the introduction to Section 3, noting that they are generally not permutation-invariant; different node orderings yield different coarsening. However, we agree that including this comparison would be valuable, and we will add a comparison with other clustering methods in the revised paper.
>
> 3. Specify Matrix W
>
> Thank you for pointing this out. We will clarify this reference in the revised manuscript. For reference, W is defined on line 190, though we agree this could be made much clearer.
>
> Thank you also for your suggestions in points 4 and 5. We will explore ways to incorporate these suggestions into the revised version.
>
> Thank you once again for your detailed feedback.
>
> We will comeback to you as soon as possible with a revised version of the paper.

---

> > ### Comment · Reviewer_F6xy · 2024-11-22
> > **Questions about comments**
> >
> > Thanks for authors' kind responses. Although the revised version has not yet been published, I have some questions regarding your comments.
> >
> > 1. Regarding invariance and equivariance in your work, you mentioned: "This property does not directly result from the use of MPNNs, as MPNNs are not directly involved in the coarsening or lifting processes." However, in my understanding, the coarsening and lifting processes are based on the assignment matrix 𝐶. That is, by ensuring the equivariance of
> > 𝐶, the coarsening and lifting processes will also be equivariant. In this case, an equivariant method to obtain
> > 𝐶 is necessary, which in your paper is achieved using the MPNN. Therefore, I believe the MPNN plays a significant role in ensuring that these processes are equivariant.
> >
> > 2. Equivariance will be important for generation if you use the spanned graph as guidance during the generation process. I believe more details about the generation model and the encoding process of the spanned graph guidance in generation could help illustrate the necessity of having an equivariant spanned graph.
> >
> > 3. Regarding the comparison of graph clustering, you mentioned: "We mention these methods briefly in the introduction to Section 3, noting that they are generally not permutation-invariant; different node orderings yield different coarsening." However, graph coarsening is a well-researched topic (e.g., [1], [2]), and many of these methods achieve permutation equivariance or invariance when employing equivariant or invariant GNNs. Is there any significant difference between their graph coarsening and yours? If not, I think your manuscript should place more focus on graph generation and how the spanned graphs are incorporated into the generation process. The coarsening and spanning sections seem somewhat lengthy compared to the main topic of your paper.
> >
> > Looking forward to your revised version.
> >
> > [1] Ma Y, Wang S, Aggarwal C C, et al. Graph convolutional networks with eigenpooling[C]//Proceedings of the 25th ACM SIGKDD international conference on knowledge discovery & data mining. 2019: 723-731.
> >
> > [2] Huang Z, Zhang S, Xi C, et al. Scaling up graph neural networks via graph coarsening[C]//Proceedings of the 27th ACM SIGKDD conference on knowledge discovery & data mining. 2021: 675-684.

---

> > > ### Author Response · Authors · 2024-11-23
> > >
> > > Thank you for your constructive comments. We fully agree with your observations and are actively revising our paper along the lines you have suggested.
> > >
> > > Regarding point (1), we agree that the equivariance of $C$ is a necessary condition for the equivariance of our method, and that, permutation-invariant coarsening is indeed well-known (3). We indeed include it as a proposition in our paper because it plays a crucial role in our method, not because it represents a contribution of ours (we will also refer to the existing coarsening method).
> > >
> > > Additionally, we are improving the presentation of the generative process in accordance with your recommendation (2).
> > >
> > > We plan to post the rebuttal version of our paper on Monday. We believe that your comments and suggestions have greatly contributed to its improvement. Thank you again for your help. We will follow up with a detailed description of the modifications.

---

> ### Author Response · Authors · 2024-11-25
>
> Thank you for your insightful comments and suggestions. We did our best to address all your feedback.
>
> For the complete list of modifications, please see the general comment above. Regarding your specific feedback, we have specifically added:
>
> - Extended and clarified the explanation of the generative process.
> - Added detailed explanations on the graph lifting process during generation.
> - Clarified the link between the Propositions and our generative model.
> - Reorganized the *Related Work* section to appear before the model presentation.
> - Enhanced the motivation for our coarsening algorithm and positioned it with respect to other coarsening methods.
> - Added an experimental comparison between our clustering method and two baseline models.
> - Included the loss function explicitly.
> - Clarified $W$ in Equation 2.
> - Moved Figure 1 to clarify that we address 2D-graph generation.
>
> We are still working on improving Figure 1.
>
> We sincerely appreciate your valuable feedback, which has greatly improved our work.

---

> ### Comment · Reviewer_F6xy · 2024-11-26
> **Questions about revision**
>
> Thanks for your revision, which tackles most of my questions. According to the revised version, I still have some concerns.
>
> 1. In my first comments, the 'graph object' means the kind of the graph. According to your manuscript, in my understanding, it's only designed for 2D graph. If 3D graph is included, the equivariance and experiments will be insufficient. So I think you should mention that at first.
>
> 2. In the generation process, in my understanding, your denoising process is based on fixed nodes and only works on edge denoising. If it is true, I am not sure whether it is meaningful in practical. Because for the molecule generation, if the atoms are fixed, the molecule is generally fixed. So most of the 2D molecule generation methods, like GraphAF[1] and GraphDF[2], they will generate nodes and edges simultaneously. Could you provide some examples of applicable scenes of your method?
>
> [1] Shi C, Xu M, Zhu Z, et al. Graphaf: a flow-based autoregressive model for molecular graph generation[J]. arXiv preprint arXiv:2001.09382, 2020.
>
> [2] Luo Y, Yan K, Ji S. Graphdf: A discrete flow model for molecular graph generation[C]//International conference on machine learning. PMLR, 2021: 7192-7203.

---

> > ### Author Response · Authors · 2024-11-26
> >
> > Thank you for your feedback and new comments.
> >
> > 1. **Regarding (1):** We are indeed generating "2D" graphs, although we find this expression imprecise since graphs do not exist in Euclidean spaces. However, based on your suggestion, we will explicitly clarify in the introduction of the next revision that our work does not involve generating objects in 3D.
> >
> > 2. **Regarding (2):** We are generating both edge and node attributes. The confusion may arise from our statement \( \mathcal{V}_\mathcal{G} = \mathcal{V}_\mathcal{H} \), which simply indicates that the two graphs have the same set of nodes (i.e., the same number of nodes). However, \( \mathcal{H} \) lacks node attributes, whereas our model generates them. For example, in molecular graph generation, our model generates both bond types (edges) and atom types (nodes). We will clarify this point in the next revision.
> >
> > Once again, thank you for your thorough review and valuable feedback.

---

> > > ### Comment · Reviewer_F6xy · 2024-11-26
> > >
> > > Thanks for authors‘ clarification, which tackles most of my concerns. I decide to improve my rating.
> > > By the way, in page 4, there is an error about appendix reference.

---

> > > > ### Author Response · Authors · 2024-12-02
> > > >
> > > > Dear Reviewer,
> > > >
> > > > We corrected the error about appendix reference.
> > > >
> > > > Thank you again for your feedback, and reviewing your rating.  We sincerely appreciate your valuable feedback, which has greatly improved our work.

---

### Official Review · Reviewer_uJpL · 2024-11-02

**Soundness:** 3
**Presentation:** 1
**Contribution:** 2
**Rating:** 5
**Confidence:** 3

**Summary:**

The paper presents a new approach to graph generation based on discrete diffusion. The method addresses two challenges in graph generative modeling, (1) scalability (2) equivariance. To address these challenges, the authors propose an equivariant coarsening-lifting technique to create sparse spanning supergraphs, which they claim maintain global graph properties. This hierarchical model enables efficient generation of large graphs, and the authors validate their approach by comparing it to existing models across multiple datasets, showing improvements in both performance and speed.

**Strengths:**

1. The usage of a coarsening/lifting technique combined with the hierarchical generation process (in an equivariant manner) is innovative in my opinion.
2. The proposed technique demonstrates competitive results compared to baseline methods.
3. The authors introduce new datasets for larger graphs.
4. The authors provide code to reproduce their results.

**Weaknesses:**

1. The presentation lacks clarity, making it difficult to understand how the model operates -- see questions below.
2. The method requires training a series of generative models, one per coarsen level. This is a weakness in my opinion, since there might be graphs that require a lot of coarsen levels.
3. Although the model shows good results on large graphs, the comparisons with baseline models are somewhat limited by generation speed constraints, leading to fewer generated samples.
4. The authors use a specific coarsening/lifting technique in their method, which might be a bottleneck in generating certain graphs. See question 2.

**Questions:**

1. Do the authors have any insights on how graph size affects the performance of generation? Specifically, is it more challenging to approximate the probability distribution for larger graphs compared to smaller ones?
2. Do the authors know how to quantify the bottleneck that comes from the fact that a particular coarsening/lifting procedure is used?
3. Does the number of nodes remain fixed throughout the entire training/sampling process? From my understanding, the diffusion process is trained independently at each coarsening level. At what point during training/generation does the number of nodes change?
4. What is the number of coarse levels (L) used in the experiments, and how does it affect performance?

---

> ### Author Response · Authors · 2024-11-13
>
> Thank you for your review, feedback, and questions. We will first address the points raised under "weaknesses" and then respond to your questions.
>
> Weaknesses:
>
> 1. Clarity
>
> We are committed to enhancing the clarity of our presentation and will submit a revised version as soon as possible to address this.
>
> 2. Number of Levels
>
> We will clarify this aspect in the manuscript. In short, all graphs in our dataset use the same number of levels, with a maximum of 4 levels in our experiments. We consider this number to be manageable and not "a lot."  We show experimentally that our method is efficient in terms of generation time. In the upcoming version of the paper, we will add indication regarding the training time.  In the light of the generative performance it allows, we do not see in what sense the hierarchical strategy would be a weakness of our model.
>
> 3. Generation Speed
>
> The generation speed is largely limited by the baseline model (specifically SparseDiff) rather than our model. Additionally, we provide results with larger sample sizes for our model in the appendix. This limitation applies only to the experiments on two specific datasets.
>
> 4. Information Loss in Coarsening/Lifting
>
> While coarsening and lifting inevitably result in some information loss, since the lifted graph is derived solely from the coarse one, we quantify this loss using spectral distance (see Section 3). However, this loss does not impact the model’s ability to generate certain graphs because:
>         1. The original graph G remains a (spanning) subgraph of the lifted graph H.
>         2. The lifted graph is used only as conditioning for the generative model, allowing it to produce any (spanning) subgraph of H.
>
> Questions:
>
> 1. Modeling large graphs
> In general, this question is challenging to answer definitively, as the difficulty of modeling a (graph) distribution primarily depends on the characteristics of the distribution itself. However, our model suggests that progressively capturing finer and larger graph structures is more manageable than attempting to model the entire graph at once. It’s an interesting question though, and we would be happy to hear your perspective on it.
>
> 2. Information Loss
>
> Please refer to point 4 in "Weaknesses" for an explanation of how information loss is handled in our model.
>
> 3. Change in the Number of Nodes
>
> During training, each level is trained independently. At each level, the number of nodes in H (the conditioning graph) and G is fixed, with the number of nodes in H determining the number in G. However, the number of nodes differs between coarsening levels. During generation, we begin by generating the coarsest level, then lift the generated graphs G^ (increasing the number of nodes at this step) to use as conditioning graphs for the subsequent level.
>
> 4. Number of Levels
>
> We used between 2 and 4 levels in our experiments (see Appendix B2). In the revised version, we will add an ablation study on the effect of the number of levels L.
>
> Thank you once again for your feedback.

---

> > ### Comment · Reviewer_uJpL · 2024-11-23
> >
> > I appreciate the authors' detailed rebuttal, and I have read the other rebuttals as well.
> >
> > The concept of hierarchical graph generation is compelling in my opinion. However, I still find that the paper's presentation lacks the level of clarity and motivation expected.
> >
> > Furthermore, despite not significantly impacting the generative performance in the experiments provided (as the authors mentioned in the rebuttal), I continue to see the independent training of each level as a limitation of the proposed approach.
> >
> > Considering these factors, I will maintain my current score.

---

> > > ### Author Response · Authors · 2024-11-25
> > >
> > > Thank you for your insightful comments and suggestions. We did our best to address all your feedback.
> > >
> > > Please note that we had not edited our paper before your last comment. You may want to look again to review the modifications.
> > >
> > > For the complete list of modifications, please see the general comment above. Regarding your specific feedback, we have specifically added:
> > >
> > > - Extended and clarified the explanation of the generative process.
> > > - Added detailed explanations on the graph lifting process during generation.
> > > - Clarified the link between the Propositions and our generative model.
> > > - Further motivated the hierarchical approach.
> > > - Enhanced the motivation for our coarsening algorithm and positioned it with respect to other coarsening methods.
> > > - Included an ablation study on the number of levels.
> > >
> > > We are still working on improving Figure 1.
> > >
> > > Regarding the independent training of each level, we emphasize that this multi-level approach offers several benefits, including:
> > > 1. Accelerating the overall training and generation process (see training and generation times, Section 5),
> > > 2. Enabling generation on large graphs, and
> > > 3. Significantly improving generative performance compared to the few available baseline methods.
> > >
> > > Moreover, nothing prevents us from training a single conditional model parameterized by the level, similar to approaches used in other models such as diffusion. However, we leave this as future work.
> > >
> > > We sincerely appreciate your valuable feedback, which has greatly improved our work.

---

> > > ### Author Response · Authors · 2024-12-02
> > >
> > > Dear Reviewer,
> > >
> > > We kindly draw your attention to the fact that we have made significant updates to our article since your last comment.
> > >
> > > All modifications are listed directly under the article.
> > >
> > > In addition to the previous changes, and in response to your feedback, we have enabled conditional generation and included insightful experiments to demonstrate its effectiveness. Please notice that conditional generation require training a single level, which answer one of your main feedback.
> > >
> > > We sincerely appreciate your valuable feedback, which has greatly improved our work.

---

### Official Review · Reviewer_Nqur · 2024-11-04

**Soundness:** 3
**Presentation:** 2
**Contribution:** 2
**Rating:** 6
**Confidence:** 4

**Summary:**

The paper propose a hierachical graph generation models. The idea is to train a partition model to partition graphs into a multiple components, where each component corresponds to a super node. The partition model helps identify components that edges mostly reside within. Then a diffusion model is applied on the components individually to avoid considering all N^2 node pair when formulating the generative models. The method claims to improve sampling speed since the number of variables to be modeled decreases.

**Strengths:**

1. The propose method has great intuition and the methods used to address graph shrinking are reasonable and convincing

2. The derivation is rigorous and I didn't spot any obvious error.

3. Experiment result seems to be superior over baselines.

**Weaknesses:**

1. The illustrative figures are a little confusing, I am not sure how the graph is lifted during the generation. Clarity need to be improved.

2. Followup question on 1. In the paper it mentions the modeling only happens intra-cluster edges, are you not considering inter-cluster edges? If so, what's the limitation of the assumption.

3. Typo in line 456: DGSS -> GDSS.

4. Metrics used in experiments are outdated, for example, MMD from GraphRNN are still used in here. I suggest using more metrics that better reflects the generated graph statistics.

**Questions:**

See weakness.

---

> ### Author Response · Authors · 2024-11-13
>
> Thank you for your thorough review and valuable feedback.
>
> 1. Presentation and Figures
>
> We will work on enhancing the overall presentation of our paper, including the clarity and quality of the figures, in the upcoming revised version.
>
> 2. Edge Modeling
>
> We model both intra- and inter-cluster edges. If there is a particular section that led to the impression that we only model intra-cluster edges, we would greatly appreciate your guidance, as this will help us improve the clarity of our description in that part of the text.
>
> 3. Typo
>
> Thank you for catching this typo; we will correct it in the revised version.
>
> 4. Metrics
>
> The metrics we use are indeed from GraphRNN. While we understand your concern, we believe these metrics remain relevant. Most recent models (such as [1] and [2]) continue to use them, and, to the best of our knowledge, no significant issues have been raised against them. If you have suggestions for alternative or improved metrics, we would be eager to consider them and would appreciate any specific recommendations.
>
> Thank you again for your insights, which are helping us to further strengthen our work.
>
>
>
> [1] Efficient and degree-guided graph generation via discrete diffusion modeling, 2023
>
> [1] Sparse training of discrete diffusion models for graph generation, 2024

---

> ### Author Response · Authors · 2024-11-25
>
> Thank you for your insightful comments and suggestions. We did our best to address all your feedback.
>
> For the complete list of modifications, please see the general comment above. Regarding your specific feedback, we have:
>
> - Answered your comment about the metrics (see previous comment).
> - Included additional motivation for the hierarchical approach (Section 2.2, revised version).
> - Added visualizations of preprocessed and generated graphs.
> - Corrected the typo (GDSS).
>
> We are still working on improving Figure 1.
>
> We sincerely appreciate your valuable feedback, which has greatly improved our work.

---

> ### Author Response · Authors · 2024-12-02
>
> Dear Reviewer,
>
> We kindly draw your attention to the fact that we have made significant updates to our article since your last comment.
>
> All modifications are listed directly under the article.
>
> In addition to the previous changes, we have improved illustrative figures, enabled conditional generation and included insightful experiments to demonstrate its effectiveness.
>
> We sincerely appreciate your valuable feedback, which has greatly improved our work.

---

> > ### Comment · Reviewer_Nqur · 2024-12-03
> > **final response**
> >
> > Upon reviewing the response, I will maintain my score.

---

### Official Review · Reviewer_nwTL · 2024-11-04

**Soundness:** 3
**Presentation:** 2
**Contribution:** 2
**Rating:** 5
**Confidence:** 3

**Summary:**

The paper introduces a hierarchical equivariant generative model for graphs using a graph coarsening-lifting method and discrete diffusion. This approach aims to address challenges including scalability and node permutation equivariance. The authors claim significant improvements in generating large-scale graphs with efficient training and inference. Empirical validation is provided through experiments on standard benchmarks and newly proposed large datasets.

**Strengths:**

The proposed coarsening-lifting method preserves global graph properties, enabling efficient generation of large graphs while maintaining node permutation equivariance. The empirical results demonstrate the method’s strong performance on various datasets, including newly introduced large-scale graph datasets.

**Weaknesses:**

1. While the hierarchical framework is compelling, training separate generative models at multiple levels increases overall complexity and may pose challenges in terms of training and hyperparameter tuning. The partitioning method also relies on training a GNN and there are no experiments verifying if this is more effective than simpler partitioning methods.
2. The descriptions of the proposed method seem quite vague and informal, which make the paper generally hard to follow. For example, how to formally define global graph information, why equivariant models can not capture that, what is the formal definition of spanning supergraph (definition in proposition 1 looks like a tautology, i.e. for any graphs $G$ and $G'$ we have $G = G' + (G\backslash G')$), etc.
3. There is no ablation study on individual components of the proposed method (such as the architectural choices, the partitioning method) or discussion about hyperparameters (such as how $L$ and diffusion steps affect the performance and efficiency).
4. The novelty seems limited given existing permutation equivariant discrete generative models for graphs [1,2]. The authors might want to provide more detailed comparisons.

[1] Efficient and degree-guided graph generation via discrete diffusion modeling, 2023

[12 Sparse training of discrete diffusion models for graph generation, 2024

**Questions:**

See above

---

> ### Author Response · Authors · 2024-11-13
>
> We thank you for your thorough review and feedbacks. We address each point raised under "weaknesses."
> 1. Multilevel and Hyperparameters
>
> We recognize the limitation mentioned but believe it should not be overstated. Most hyperparameters can be applied across various graph levels, and the model's performance is not highly sensitive to fine-tuning of these parameters.
> Regarding the lack of comparison with other methods, we will provide a revised version of our manuscript that includes a section discussing and comparing alternative coarsening methods. If you have any specific methods to suggest for inclusion, we would greatly appreciate your input.
>
> 2. Formal Description of the Model
>
> We will refine the presentation to improve clarity and will address this in the revised paper.
>
> Concerning the definition of a spanning supergraph, a formal definition is given on line 100. In Proposition 1, please note that $\mathcal{S}$, $\mathcal{E}_G$​, and $\mathcal{E}_H​$ refer to sets of edges, not graphs. We will clarify this distinction and provide an updated version.
>
> 3. Ablation Study
>
> The comparison between our hierarchical method (HEDD) and our baseline (DiscDiff), using the same architecture and hyperparameters, serves as an ablation of our hierarchical approach.
>        Additionally, we plan to include an experiment that ablates the number of levels in our model.
>
> 4. Comparison with Other Models
>
> The two models mentioned are discussed in Section 4 (Related Works) and serve as baselines in our experiments. Could you please clarify what you mean by a more detailed comparison? Are you suggesting additional experiments, or would you prefer a more detailed discussion of the two methods with an emphasis on differences from our model?
>
> Thank you once again for your detailed feedback.
>
> We will comeback to you as soon as possible with a revised version of the paper including the modification.

---

> ### Author Response · Authors · 2024-11-25
>
> Thank you for your insightful comments and suggestions. We did our best to address all your feedback.
>
> Please, for the complete list of the modifications, see the general comment above. Regarding, your feedback, we have specifically added:
>
> - Polished the presentation and added formalism.
> - Better defined a spanning supergraph (see Definition 1, revised version).
> - An experimental comparison between our clustering method and two baseline models.
> - An ablation study on the number of levels.
> - A detailed comparison with SparseDiff and EDGE.
>
> We sincerely appreciate your valuable feedback, which has greatly improved our work.

---

> ### Author Response · Authors · 2024-12-02
>
> Dear Reviewer,
>
> We kindly draw your attention to the fact that we have made significant updates to our article since your last comment.
>
> All modifications are listed directly under the article.
>
> In addition to the previous changes, we have enabled conditional generation and included insightful experiments to demonstrate its effectiveness.
>
> We sincerely appreciate your valuable feedback, which has greatly improved our work.

---

### Official Review · Reviewer_ZMRK · 2024-11-04

**Soundness:** 3
**Presentation:** 2
**Contribution:** 3
**Rating:** 6
**Confidence:** 3

**Summary:**

The authors of this paper introduce a hierarchical approach for graph generation using graph coarsening and discrete diffusion. The authors introduce a novel "coarsening-lifting" method that creates spanning supergraphs to model global graph properties at each hierarchical level, improving scalability and capturing global structure. They evaluate their method against other graph generative models and achieve promising results on multiple datasets.

**Strengths:**

- One of the strongest points of the paper is the scalability of the proposed methods. Usually, graph generators struggle with large graphs, as they have quadratic complexity. By leveraging sparse supergraphs for message-passing, the model reduces this complexity, achieving lower generation time for the graphs.

-   The experimental results are strong, outperforming previous several state-of-the-art methods. The model demonstrates better performance, faster graph generation, and lower memory consumption, which is especially apparent with larger datasets.

- Hierarchical graph generation is an interesting and important research direction, as it can better capture the hierarchical patterns in various graph datasets.

**Weaknesses:**

- A weak point of the paper is that it doesn’t sufficiently motivate the hierarchical generation approach beyond its computational efficiency. While reducing computation costs is a valuable advantage, hierarchical generation could offer additional benefits, such as enhanced model interpretability or improved capture of multi-scale graph structures. The paper would be stronger if it discussed these potential benefits in more detail, especially how hierarchical modeling might help in applications with hierarchical graph patterns.

- Even if the paper acknowledges the lack of conditional generation, I think it is important to include some preliminary results or ideas on how to extend the current method with conditioning. In practical applications, generating graphs conditioned on specific properties (e.g., molecule properties) is often necessary.

- Another weak point is the lack of visualizations for the generated graphs. Visual comparisons between generated and real graphs would further improve the evaluation process. Graph visualizations could also help illustrate how well the hierarchical coarsening-lifting process preserves essential features at each level.

- The training time is missing for the results, and only the generation time is included.

**Questions:**

- Could the authors elaborate on the use for hierarchical generation, aside from computational cost reduction? Specifically, how the proposed method can better model and generate graph structures with hierarchical patterns?

- How the authors would extend their method to conditional graph generation?

-  Could the authors provide visual comparisons between generated and real graphs?

-  Could the authors provide training time for their method and the baselines across different datasets?

---

> ### Author Response · Authors · 2024-11-13
>
> We thank you for your thorough review and thoughtful feedback. Below, we respond to each of the points raised under "weaknesses".
>
> 1. Benefits of Hierarchical Generation
>
> Our intention was indeed to show not only how our model helps scale graph generation to larger graphs but also how the hierarchical structure progressively captures global graph properties. For instance, this is the intention behind our theoretical analysis regarding spectral properties. However, we acknowledge that we can improve the presentation to emphasis the other benefits of our hierarchical model.
>
> To address this, we will:
>   - Extend our analysis to further illustrate the advantages of the hierarchical model.
>   - Improve the presentation to emphasize these benefits more effectively.
>
> 2. Conditional Generation
>
> As we acknowledge in our paper, conditional generation is indeed important. Currently, our model does not propose a specific module for incorporating global graph properties; however, we  can include global features by concatenating the same global feature vector to each input node attribute. We will expand on this idea in the paper to clarify how global features can be incorporated.
>
> However, adding experiment could be complicated in practice. The ZINC dataset does not include molecular properties, but we will explore if we can compute some properties using Rdkit. The QM9 dataset, which includes several molecular properties and consists of small graphs (up to 9 heavy atoms), is too small and therefore not relevant for our method.
>
> If you have specific suggestions for datasets and experiments that would best demonstrate conditional generation with our model, we would be happy to consider them.
>
> 3. Visualizations
>
> We agree that visualization would strengthen our paper, and we will include visualizations comparing real and generated graphs. We also plan to add visualizations of the coarsening/lifting procedure on real graphs to illustrate the model’s hierarchical processes.
>
> 4. Training Time
>
> We thank you for this suggestion as well. We plan to include the training time, presenting it as the average time per epoch for a fixed batch size. Note that we have already provided the training time for the partitioning process in Table 1.
>
> Thank you once again for your detailed feedback, which is very helpful. Please let us know if you have any further suggestions.

---

> ### Author Response · Authors · 2024-11-25
>
> Thank you for your insightful comments and suggestions. We did our best to address all your feedback.
>
> Specifically, we have added:
>
> - Additional motivation for the hierarchical approach (Section 2.2, revised version).
> - An explanation of how our model can be used for conditional generation.
> - Visualizations of preprocessed and generated graphs.
> - Training times for the experiments.
>
> We sincerely appreciate your valuable feedback, which has greatly improved our work.

---

> ### Author Response · Authors · 2024-12-02
>
> Dear Reviewer,
>
> We kindly draw your attention to the fact that we have made significant updates to our article since your last comment.
>
> All modifications are listed directly under the article.
>
> In addition to the previous changes, and in response to your feedback, we have enabled conditional generation and included insightful experiments to demonstrate its effectiveness.

---

### Author Response · Authors · 2024-11-25

Dear Reviewers,

We sincerely appreciate your constructive suggestions and would like to thank you warmly for your thorough and insightful comments and questions. We strongly believe your feedback has significantly improved our article.

**We have now updated our article** and kindly encourage you to take a look at the revised version.

Below is a summary of the changes we have made:

- **Paper Presentation and Clarity**:
  - Extended and clarified the explanation of the generative process.
  - Added detailed explanations on the graph lifting process during generation.
  - Clarified the link between the Propositions and our generative model.
  - Further motivated the hierarchical approach.
  - Reorganized the *Related Work* section to appear before the model presentation.
  - Enhanced the motivation for our coarsening algorithm and positioned it with respect to other coarsening methods.

- **Additions Following Your Suggestions**:
  - An ablation study on the number of levels.
  - An experimental comparison between our clustering method and two baseline models.
  - An explanation of how to use our model for conditional generation.
  - Visualizations of preprocessed and generated graphs.
  - Training times for our experiments.
  - A detailed comparison with two other sparse equivariant models, SparseDiff and EDGE.
  - Minor corrections, including typos and clarification of Equation 2.

We are also working on further improving the paper with the following planned updates:

- Adding an experiment on conditional graph generation.
- Enhancing the readability of Figure 1.

Once again, we thank you for your valuable feedback, which helped us in improving the quality of our work.

Best regards,
The authors

---

### Author Response · Authors · 2024-11-28

Dear Reviewers,

**We have updated our article** and kindly encourage you to review the revised version.

In addition to the previous updates, we have:

- Included a new experiment on conditional graph generation.
- Improved Figure 1.
- Added additional visualizations, including those for the conditional model.
- Clarified the type of graphs we generate (2D) and provided additional details on the generation of graph attributes, such as bond types and atom types.

We take this opportunity to highlight the key contributions of our paper:

- We propose a new hierarchical generative model that scales to large graphs and, to the best of our knowledge, is the first graph generative approach that is both hierarchical and equivariant.
- Our model is faster in both training and generation while achieving better performance in capturing graph distributions compared to baselines.
- Our method inherently provides a conditional generative framework for structure-guided graph generation, enabling effective spectrum-conditioned generation.
- We introduce a novel graph coarsening method to produce minimal spanning supergraphs, which serve as conditioning structures and as sparse frameworks for generation.
- We prove that our coarsening method maximizes the sparsity of the conditioned structure while preserving important spectral properties. Experimentally, we demonstrate that our approach outperforms existing coarsening baseline methods.
- We present new datasets with more instances and larger graphs, enabling more comprehensive evaluation of graph generative models.

Although the revision period has ended, we would greatly appreciate your feedback on the revised version if you have not yet provided it.

Once again, thank you for your thorough and insightful comments and questions. We firmly believe your feedback has significantly improved our article.

**Best regards,**
The authors

---

### Meta-Review · Area_Chair_eV14 · 2024-12-20

**Metareview:**

1, This paper presents a hierarchical graph generation model for (2D) graph generation, addressing two issues, scalability and equivariance. An equivariant coarsening-lifting technique is developed to create sparse spanning supergraphs,  to model global graph properties at each hierarchical level. Although the idea of hierarchical generation is not new enough, combining the sparse-spanning supergraphs delivers some value to the community. However, many concerns remain, including the weak motivation of the hierarchical fashion especially considering the efficiency issue, limited novelty (especially the permutation equivariance has been extensively studied, and the used MPNN seems to naturally achieve this, the lack of technical details, the lack of necessary condition generation results, independent training of each hierarchical level, etc. Also, the authors failed to revise the manuscript in a timely manner, and many suggested comments are not reflected in the current version. Given these considerations, I feel the paper is not ready to publish in its current state.

**Additional Comments On Reviewer Discussion:**

The authors worked hard to address the concerns of the reviewers, but some significant concerns remained, including the weak motivation, limited novelty, the claim on the equivariance, and the lack of conditional generation experiments. Additionally, during the discussion, the authors posted responses quickly, but the revised paper was uploaded just a few days before the deadline, which I believe hindered the communications between the reviewers and authors. I suggest the authors pay attention to this for rebuttal in future venues like ICML and NeurIPS.

---

### Decision · Program_Chairs · 2025-01-22

Reject